# Simulational Tests of the Rouse Model

**DOI:** 10.3390/polym15122615

**Published:** 2023-06-08

**Authors:** George David Joseph Phillies

**Affiliations:** Department of Physics, Worcester Polytechnic Institute, Worcester, MA 01609-2280, USA; phillies@4liberty.net; Tel.: +1-508-754-1859

**Keywords:** polymer dynamics, molecular dynamics, Brownian dynamics, Rouse model, Kirkwood–Riseman model, computer simulation, Rouse modes

## Abstract

An extensive review of literature simulations of quiescent polymer melts is given, considering results that test aspects of the Rouse model in the melt. We focus on Rouse model predictions for the mean-square amplitudes $\langle {\left(X_{p}\left(0\right)\right)}^{2}\rangle$ and time correlation functions $\langle X_{p}\left(0\right)X_{p}\left(t\right)\rangle$ of the Rouse mode $X_{p}\left(t\right)$. The simulations conclusively demonstrate that the Rouse model is invalid in polymer melts. In particular, and contrary to the Rouse model, (i) mean-square Rouse mode amplitudes $\langle {\left(X_{p}\left(0\right)\right)}^{2}\rangle$ do not scale as $\sin^{-2}\left(p\pi /2N\right)$, *N* being the number of beads in the polymer. For small *p* (say, p≤3) $\langle {\left(X_{p}\left(0\right)\right)}^{2}\rangle$ scales with *p* as $p^{-2}$; for larger *p*, it scales as $p^{-3}$. (ii) Rouse mode time correlation functions $\langle X_{p}\left(t\right)X_{p}\left(0\right)\rangle$ do not decay with time as exponentials; they instead decay as stretched exponentials $\exp(-\alpha t^{\beta })$. β depends on *p*, typically with a minimum near N/2 or N/4. (iii) Polymer bead displacements are not described by independent Gaussian random processes. (iv) For p≠q, $\langle X_{p}\left(t\right)X_{q}\left(0\right)\rangle$ is sometimes non-zero. (v) The response of a polymer coil to a shear flow is a rotation, not the affine deformation predicted by Rouse. We also briefly consider the Kirkwood–Riseman polymer model.

## 1. Introduction

The Rouse model [1] is the foundational basis for much of modern polymer physics. As seen in seminal early works [2,3], definitive research monographs [4], and textbooks, the Rouse model for an ideal polymer is said to describe polymer motion in lower-molecular weight polymer melts to describe short-time and ’within the tube’ polymer motion of high-molecular-weight polymer melts, and, after inserting hydrodynamic interactions [5], to describe the motion of polymers in dilute solution.

Rouse [1] proposed a bead-and-spring model for an isolated polymer chain in solution, with individual beads subjected to uncorrelated hydrodynamic and random thermal forces, a harmonic force between pairs of bonded beads, and sufficient hydrodynamic drag that bead inertia could be neglected. Rouse’s model leads to a set of coupled linear differential equations for the motions of the beads. The solutions to these differential equations are the Rouse modes, whose mean-square amplitudes are predicted by the model, and whose amplitudes on the average decay exponentially with time. From the mode behavior and various mechanical assumptions, the Rouse model predicts the polymer’s diffusion coefficient, the polymer’s contribution to the solution viscosity, and the motion of the polymer’s end-to-end vector.

There are other models for polymer dynamics and polymers in solution. In particular, Kirkwood and Riseman [6] proposed a bead-and-spring polymer model, whose description of the important polymer motions is entirely opposite to that of Rouse. The author of Refs. [7,8] extended the Kirkwood–Riseman model to concentrated polymer solutions.

One path to testing these models is the use of computer simulations. Even limiting our review to molecular dynamics methods, one readily identifies well over a thousand articles that might be included in a comprehensive review of simulations of polymer dynamics, not counting research approaching engineering applications of simulations. Simulations using Monte Carlo methods, in which momentum variables are suppressed, are not relevant to tests of dynamics. In addition, also not considered here, there is an extremely large but non-communicating research field on computer simulations of the dynamics of biopolymers and structures, e.g., complete ribosomes [9]. Finally, going back two-thirds of a century, applications of computational methods to a Rouse-like model (albeit with a much more elaborate system Hamiltonian) of simple organic molecules were used to interpret their infrared and Raman vibrational frequencies.

There is disagreement in the literature as to the validity of the Rouse model and its utility for describing polymer dynamics. The original simulation of a single Rouse chain by Grest and Kremer [10] did confirm the model. For melts, matters are more complicated. Tsolou et al. [11] say of their simulations that “*…The Rouse Theory is found to provide a satisfactory description of the simulation findings, especially for rings with chain length between C*_50_* and C*_170_*…*”. Roh et al. [12] conclude that “*…the structure and relaxation of the unentangled short-chain-branched ring and linear melt systems can be reasonably well characterized with the Rouse model, regardless of the short branches…*”. Tsalikis et al. [13] conclude that “*…Overall, our MD simulations have demonstrated the validity of the Rouse model for the dynamics of the simulated PEO ring melts…*”. Kopf et al. [14] report that the Rouse model remained accurate in blends of light and heavy polymer chains.

Some authors describe the Rouse and reptation models as assumptions. For example, Colmenero [15] observes that “*Nowadays it is generally assumed that the Rouse model provides a suitable description of chain dynamics of unentangled polymer melts…*”. Colmenero continues by noting that the reptation model assumes the validity of the Rouse model at short times. He then notes the Rouse model’s failures and the model predictions that the failures invalidate.

On the other hand, in his review *Viscoelasticity and Molecular Rheology* in *Polymer Science: A Comprehensive Reference* [16], Likhtman observes that “*We note that often models are studied by theoreticians just because they are analytically solvable and used by experimentalists because of availability of analytic solutions…*”, leading to his conclusion “*This coupling* [GP: between Rouse modes; see below] *suggests that the Rouse mode description is not very useful for entangled polymers*”. Kalathi et al. [17] note Likhtman’s observation, yet use a Rouse mode analysis in their work, saying as a sensible defense that experimentalists “*…still tend to model chain dynamics in the language of the Rouse model. Understanding experimental results therefore requires us to analyze the simulations in the same manner*”.

There are two fundamental sorts of tests of any model of polymer dynamics, namely *direct* tests and *inferential* tests. In direct tests, one examines what the model actually says about polymer motions. The Rouse modes either do or do not have the behavior predicted by the model. In inferential tests, one compares the model’s predictions for measurable parameters with the experimental behavior of the parameters. A well-known inferential test of the Rouse model is the dependence of the melt viscosity on the polymer molecular weight *M*. The observed molecular-weight-dependence for melts of lower-molecular-weight polymers agrees with calculations based on the model, leading one to observe that the model agrees with the experiment. The power of inferential tests can be overstated. The observed *M*-dependence would only be a demonstration that polymer dynamics are Rouse-like if the prediction could be shown to be unique, i.e., the molecular weight dependence would only show that Rouse dynamics are correct if one could show that no fundamentally different model makes the same prediction. As it happens, there is actually a fundamentally different model of polymer dynamics, the Kirkwood–Riseman model with hydrodynamics suppressed, that predicts the same *M*-dependence, so the inferential test supplies a necessary, but not sufficient, test of the validity of the Rouse model.

This review focuses on simulations of quiescent polymer melts that make direct tests of the Rouse modes, their amplitudes, and the time dependence of their relaxations. We identified a substantial number of these simulations. To the limited extent that we found simulational results that did not agree with our conclusions, we were careful to include those results. There are also extensive simulations of polymer melts and solutions experiencing shear. Those simulations are largely beyond the scope of this review.

In the following, Section 2 of the paper describes the Rouse and Kirkwood–Riseman models. These models are familiar from the literature, so we limit ourselves to descriptions of these models that are adequate to establish notation and terminology. Section 3 presents some general tests for the models. The following Section 4 describes seriatim the simulation studies being reviewed and what they reveal. Almost all of these works were concerned with polymer melts. The points at which the Kirkwood–Riseman model differs from the Rouse model correspond to questions that have rarely been asked in simulations; one study (which happens to support the Kirkwood–Riseman model) is noted. An extended discussion unifying these results follows as a further section, Section 5. Some readers may prefer to skip to Section 5 and then return to read fine details in Section 4. The conclusions in Section 6 close the paper.

## 2. Models of Polymer Dynamics

Here, we treat the Rouse and Kirkwood–Riseman models. This review is primarily about the widely used Rouse model, not the relatively rarely encountered Kirkwood–Riseman model, which has had few simulational tests.

The Rouse and Kirkwood–Riseman models for polymer dynamics both describe a polymer chain as a line of *N* hydrodynamically active beads, labeled {0,1,2,…,N−1} in order along the chain, linked by hydrodynamically inert Hookean springs. The beads have Cartesian coordinates $\left(r_{0},r_{1},\ldots r_{N-1}\right)$ and hydrodynamic drag coefficients $\left(f_{0},f_{1},\ldots ,f_{N-1}\right)$. The springs, with force constants $k_{i}$, serve to control the average distance between bonded pairs of beads. The hydrodynamic force on a bead *i* is taken to be linear in the velocity of the bead with respect to the solvent, namely
(1)$$F_{iH}=f_{i}\left(v_{i}-{\dot{r}}_{i}\right).$$

Here, ${\dot{r}}_{i}$ is the velocity of bead *i*, while $v_{i}$ is the velocity that the fluid would have had, at the location of bead *i*, if the bead were absent.

How do the model images of beads correspond to the detailed chemical structure of real polymers? Kirkwood and Riseman are precise; the polymer beads are the monomers along the polymer chain. In their model, the approximation that the distance between two remote beads has a Gaussian distribution of values is extrapolated to include distances between beads that are nearby along the polymer chain. On the other hand, in the Rouse model, the polymer chain is effectively coarse grained into a series of segments, the distances between different segments having a Gaussian distribution. It is often assumed that the Rouse model’s segments are each a Kuhn length in extent.

In most treatments, the bead drag coefficients $f_{i}$ and force constants $k_{i}$ are taken to have common values *f* and *k*, respectively. *f* is taken to be large enough that the bead motions are overdamped on time scales of interest so that bead inertia can be neglected. In a few studies, simulations in which some beads have large sizes have given physically interesting results. When the solvent exerts a force on a bead, from Newton’s Third Law, the bead exerts an equal and opposite force on the solvent. In the Zimm and Kirkwood–Riseman models, but not the Rouse model, the forces that the beads exert on the solvent create solvent flows that perturb the motion of the solvent around each of the other beads. These perturbations are the bead–bead hydrodynamic interactions, described in both the Zimm and the Kirkwood–Riseman models by the Oseen tensor.

The polymer coils are part of a thermal system. Corresponding to the frictional forces of Equation (Equation 1), the fluctuation–dissipation theorem guarantees that there must on each bead be a fluctuating thermal force $F_{i}\left(t\right)$, the thermal forces serving to maintain the temperature of the system. In the Rouse model, there are no hydrodynamic interactions between the beads, so the $F_{i}\left(t\right)$ on different beads are uncorrelated. In the presence of hydrodynamic interactions, the fluctuating thermal forces on different beads must necessarily have cross correlations. These considerations lead to equations of motion for the beads, in the form of a set of coupled linear differential equations with constant coefficients. The end beads of linear chains are special cases. For the other beads, one has
(2)$$f{\dot{r}}_{i}=fv_{i}+k\left(r_{i+1}+r_{i-1}-2r_{i}\right)+F_{i}\left(t\right)$$

The solution of the Rouse model begins with the Rouse modes, which are a discrete Fourier transform between an index *i* that labels the position coordinates $r_{i}\left(t\right)$ of the beads and an index *p* that labels the *N* Rouse modes $X_{p}\left(t\right)$. The Rouse modes, like the position coordinates, form a complete orthogonal set of coordinates that between them specify the positions of the beads in a polymer chain. For a linear chain, $X_{p}$ is related to $r_{i}\left(t\right)$ by
(3)$$X_{p}\left(t\right)=\frac{1}{N}\sum_{i=0}^{N-1}r_{i}\left(t\right)\cos\left(\frac{p\pi (i+1/2)}{N}\right).$$
with an inverse
(4)$$r_{i}\left(t\right)=X_{0}\left(t\right)+2\sum_{p=1}^{N-1}X_{p}\left(t\right)\cos\left(\frac{p\pi (i+1/2)}{N}\right).$$

The Cartesian representation of the $p^{th}$ normal mode is found from Equation (Equation 4) by setting one Rouse coordinate $X_{p}\left(t\right)=1$ and setting all the other Rouse coordinates to zero.

The $X_{p}\left(t\right)$ are the normal modes of the Rouse model. In the Rouse model, we have the following:The random forces and corresponding thermal displacements of the beads in a polymer chain are described by independent Gaussian random processes.The $X_{p}\left(t\right)$ are normal modes so that $\langle X_{p}\left(0\right)X_{q}\left(t\right)\rangle =0$ if p≠q. (Being a normal coordinate and being a normal mode are not the same. One is a statement about a mathematical linear transformation; the other is a statement about the system dynamics.)The ensemble-average mean-square amplitude $\langle {\left(X_{p}\left(0\right)\right)}^{2}\rangle$ is determined by *p*, *N*, and material variables.The correlation function $\langle X_{p}\left(0\right)X_{p}\left(t\right)\rangle$ of each mode decays as a pure exponential $\langle {\left(X_{p}\left(0\right)\right)}^{2}\rangle \exp\left(-t/\tau_{p}\right)$, $\tau_{p}$ being determined by *p*, *N*, and material variables.For each mode, the bead velocities $dr_{i}\left(t\right)/dt$ are always directed exactly opposite to $r_{i}\left(t\right)$, i.e., they are always directed at the chain center-of-mass. When the chain is at rest, all beads are at the same location, so that the chain radius is zero.Under shear, a polymer coil responds via an affine deformation. At low frequencies, the beads are stationary, because the hydrodynamic shear force and the force due to the bonds cancel each other.

In particular,
(5)$$\tau_{p}^{-1}=\frac{12k_{B}T}{fb^{2}}\sin^{2}\left(\frac{p\pi }{2N}\right)\approx \left(\frac{3\pi^{2}k_{B}T}{fb^{2}}\right)\frac{p^{2}}{N^{2}}$$
and
(6)$$\langle {\left(X_{p}\left(0\right)\right)}^{2}\rangle =\frac{b^{2}}{8N}\sin^{-2}\left(\frac{\pi p}{2N}\right)\approx \frac{Nb^{2}}{2\pi^{2}p^{2}},$$
where *b* is the root-mean-square average distance between two neighboring beads, $k_{B}$ is Boltzmann’s constant, *T* is the absolute temperature, and the two approximations are valid for small p/N. The $X_{p}\left(0\right)$ are predicted to have independent Gaussian random distributions, so all higher moments of $X_{p}\left(0\right)$ can be calculated from $\langle {\left(X_{p}\left(0\right)\right)}^{2}\rangle$. The statement that the modes are orthogonal at all times so that $\langle X_{p}\left(0\right)X_{q}\left(t\right)\rangle =0$ if p≠q arises from the forces in the Rouse model, and is not equivalent to the equally correct statement that the Rouse coordinates are orthogonal. It is straightforward to make a modest modification of the Rouse model such that some modes become cross correlated, namely, one applies to the molecule a time-independent external shear field [18].

From the Rouse modes, one may obtain predictions for the self-diffusion coefficient, the viscoelastic behavior, mean-square bead displacements, and the relaxation of the polymer end-to-end vector. These predictions are largely outside the scope of this review.

Many analyses begin by assuming the fundamental validity of the Rouse [1] and Zimm [5] models of polymer dynamics, at least on some time and distance scales. It is not always recalled that the Rouse and Zimm models were preceded by the Kirkwood–Riseman model [6]. When the Kirkwood–Riseman model is mentioned at all, it tends to be treated as being much the same as the Rouse and Zimm models. As a grain of truth, these three models all describe a single isolated polymer coil, not a polymer melt.

Kirkwood and Risemann discuss the motions of a polymer coil in a shear flow. The polymer has a center of mass $R_{\mathrm{CM}}\left(t\right)$. In the Kirkwood–Risemann model, the velocity of a bead *i* is
(7)$$\frac{dr_{i}\left(t\right)}{dt}=\frac{dR_{\mathrm{CM}}\left(t\right)}{dt}+\Omega \times \left(r_{i}\left(t\right)-R_{\mathrm{CM}}\left(t\right)\right)+u_{i}\left(t\right).$$

Here, $\frac{dR_{\mathrm{CM}}\left(t\right)}{dt}$ is the average translational velocity of all the beads, Ω is the polymer’s rotational velocity, and $u_{i}\left(t\right)$ is the part of the bead’s velocity due to chain internal modes. In the remainder of Kirkwood and Riseman’s calculations, chain internal modes and the velocities they create are ignored, as providing only secondary corrections. $\frac{dR_{\mathrm{CM}}\left(t\right)}{dt}$ and Ω are determined by the restriction that the system is heavily overdamped, meaning that the net force and the net torque on the polymer must, except on very short time scales, average very nearly to zero. In a quiescent fluid, one then necessarily has $\frac{dR_{\mathrm{CM}}\left(t\right)}{dt}=0$ and Ω=0. In a simple shear field $dv_{x}/dy=\mathrm{constant}$, it is generally impossible for every bead of a polymer chain to move at the velocity given by the local value of the shear field. As a result, the polymer translates at the average of the fluid speeds seen by all the beads. The polymer also rotates with torques due to the *x* and *y* components of the rotational velocity of all the beads on the average canceling. However, at any time, most beads are moving with respect to the neighboring fluid, the dissipation arising from the corresponding viscous drag being the viscosity increment created by the polymer. In the absence of a shear flow, a Kirkwood–Riseman model chain performs translational and rotational diffusion, the latter having the effect of relaxing the direction of the chain end-to-end vector.

It is generally ignored that the Kirkwood–Riseman and Rouse models give completely contradictory descriptions of how polymer coils move in solution and create viscous dissipation. Consider a linear chain having *N* beads. In all models, the polymer coil has three center-of-mass coordinates with a center-of-mass velocity corresponding to an average translational motion of the entire chain. In the Rouse model, the Rouse transformation replaces the 3N Cartesian coordinates with three center-of-mass coordinates and 3N−3 internal Rouse coordinates $X_{p\alpha }\left(t\right)$. Here, p∈(1,N−1) and α∈(x,y,z). In the Rouse model, each $X_{p\alpha }\left(t\right)$ corresponds to a Rouse mode. The 3N−3 Rouse modes have a common feature, namely that in each mode at least some of the beads move with respect to each other.

In contrast to the Rouse and Zimm models, in the Kirkwood–Riseman model, an *N*-bead polymer coil has three translational modes, describing an averaged translation of the polymer chain, and three rotational modes, describing an averaged rotation of the polymer chain. In the translational and rotational components of the bead motions, the distances between the polymer beads remain constant. There then remain 3N−6 internal modes, in which the relative positions of the beads change with time. The Rouse and Kirkwood–Riseman models thus do not agree as to how many internal modes—modes in which the beads move with respect to each other—a polymer coil has. One model says 3N−3 modes, while the other says 3N−6 modes.

It should have been, but was not, immediately apparent that the Rouse model with its 3N−3 independent internal modes is inconsistent with basic classical mechanics, in which an *N*-atom molecule can translate and rotate, and therefore has 3N−6 internal degrees of freedom. This count of the allowed number of internal modes is established with absolute certainty by experimental and theoretical studies of infrared and Raman spectroscopy. Furthermore, prominently from Raman spectroscopy of molecular crystals, rotational modes are slow relative to most vibrational modes, so it would be incorrect to propose that the discrepancy in the mode count can be hidden in a few high-frequency modes that are otherwise of no significance.

The Rouse and Kirkwood–Riseman models are also entirely opposite in their descriptions of how polymeric viscosity increments arise. Rouse assigns viscous dissipation to the polymer chain’s internal modes, which leads to forces opposing the hydrodynamic forces due to the shear field, while denying whole-body rotation in a shear field. In a shear field, in Rouse’s model, a polymer is subject to an affine deformation. Its internal forces due to the deformation lead to viscous dissipation. Kirkwood and Riseman assign viscous dissipation to whole-body rotation, while neglecting internal motions of a chain as providing only secondary corrections.

In the Rouse model, the correlation functions $\langle X_{p\alpha }\left(t\right)X_{p\alpha }\left(0\right)\rangle$ relax exponentially in time. Contrariwise, if mode relaxations are stretched exponentials in time, or have some other time dependence, then assuredly the underlying polymer dynamics are not those of the Rouse model. In each Rouse mode, most beads are displaced from their rest positions and return back to them as time advances. In the Rouse model, all beads have the same rest position; at rest, all beads are at the center of mass. Rouse modes correspond to spatially (but not temporally) oscillatory displacements having larger or smaller wavelengths. Rouse modes are not wavelets; they do not refer to fluctuations in a single localized region of a chain. The Zimm model [5] is substantially similar to the Rouse model, except that bead–bead hydrodynamic interactions at the level of the Oseen tensor are included in the calculation. These hydrodynamic interactions change how rapidly each mode relaxes, but the internal modes of the Zimm model are the same as the internal modes of the Rouse model, except for their relaxation rates.

Rouse applied his model [1] to calculate the viscosity increment created by a Rouse-model polymer. An applied shear field, with fluid velocity along the *x* axis, velocity gradient along the *y*-axis, and vorticity vector in the *z* direction, was claimed to displace polymer beads but, according to Rouse, only in the *x* direction, the direction of the velocity. Motions of the polymer beads in the *y* and *z* directions were claimed not to be affected by this shear field. In the Rouse model, a polymer chain subject to an external shear field thus performs an affine deformation as described more recently by deGennes [2], but, in the Rouse model, polymer chains in a shear field do not rotate. The forces of the polymer beads on the solvent, due to their having been displaced relative to each other by the shear, as described by Rouse modes, lead in the model to viscous dissipation.

In the Rouse model, every monomer bead is exposed to a Gaussian random force. The displacements $r_{i}\left(t\right)-r_{j}\left(0\right)$ arise from weighted linear sums of these external forces, so $r_{i}\left(t\right)-r_{j}\left(0\right)$ must, in the Rouse model, have a Gaussian random distribution, leading to
(8)$$\langle (\exp\left(ıq\cdot \left(r_{i}\left(t\right)-r_{j}\left(0\right)\right)\right)\rangle =\exp\left(-\frac{q^{2}}{6}\langle {\left(r_{i}\left(t\right)-r_{j}\left(0\right)\right)}^{2}\rangle \right)$$
and therefore
(9)$$g^{\left(1\right)}\left(q,t\right)=\sum_{i=1}^{N}\sum_{j=1}^{N}\exp\left(-\frac{q^{2}}{6}\langle {\left(r_{i}\left(t\right)-r_{j}\left(0\right)\right)}^{2}\rangle \right)$$

This is the Gaussian approximation for the intermediate structure factor $g^{\left(1\right)}\left(q,t\right)$; its time dependence is determined by the one- and two-particle time-dependent mean-square particle displacements. (Some authors denote $g^{\left(1\right)}\left(q,t\right)$ as S(q,t)). Chong and Fuchs [19] offer a demonstration that this Gaussian approximation is theoretically appropriate. We return later to the accuracy of this approximation as tested by simulations.

An interesting physical question is the possible presence of cross-correlations in Rouse mode amplitudes in physical systems, i.e., is $\langle X_{pi}\left(t\right)X_{qj}\left(0\right)\rangle$ ever non-zero for p≠q and/or i≠j? Such correlations do not exist in a Rouse-model polymer but could exist in some system that is not described by the Rouse model. A plausible general form for a cross-time-correlation function of two Rouse modes is
(10)$$\Phi_{pqij}\left(t\right)=\frac{\langle X_{pi}\left(t\right)X_{qj}\left(0\right)\rangle }{{\langle {\left(X_{pi}\left(0\right)\right)}^{2}\rangle }^{1/2}{\langle {\left(X_{qj}\left(0\right)\right)}^{2}\rangle }^{1/2}}.$$

A corresponding equation, not the most general one, for cross correlations between the mean-square amplitudes of two Rouse modes would be
(11)$$\Phi_{pqij}^{\left(4\right)}\left(t\right)=\frac{\langle |X_{pi}\left(t\right){|}^{2}|X_{qj}\left(0\right){|}^{2}\rangle }{\langle {\left(X_{pi}\left(0\right)\right)}^{2}\rangle \langle {\left(X_{qj}\left(0\right)\right)}^{2}\rangle }.$$

The elaborate normalization seen here is invoked because even a variable whose typical size is small can still make a strong contribution, relative to its size, to the variable with which it is correlated. From the above definition, $\Phi_{ppii}\left(0\right)=1$; a variable is perfectly correlated with itself. In an equilibrium, nonchiral fluid, from reflection symmetry, $\Phi_{pqij}\left(0\right)=0$ if i≠j. For an isolated Rouse chain *in a fluid with shear*, this author previously showed from simulations that $\Phi_{ppij}\left(t\right)\neq 0$ can occur. In the presence of shear, one finds cross correlations for p=q and i≠j [18].

In many cases, $\langle X_{pi}\left(t\right)X_{pi}\left(0\right)\rangle$ is found simulationally to decay as a stretched exponential in time, i.e.,
(12)$$\langle X_{pi}\left(t\right)X_{pi}\left(0\right)\rangle =\langle {\left(X_{pi}\left(0\right)\right)}^{2}\rangle \exp\left(-{\left(\frac{t}{\tau }\right)}^{\beta }\right)\equiv \langle {\left(X_{pi}\left(0\right)\right)}^{2}\rangle \exp\left(-\alpha t^{\beta }\right)$$

Some authors proposed that a characteristic time $\tau_{p}$ may be extracted from this form, by analogy with the simpler integral
(13)$$\tau =\int_{0}^{\infty }dt\;\exp\left(-t/\tau \right),$$
namely
(14)$$\tau_{p}=\int_{0}^{\infty }\;dt\;\exp\left({\left(-\frac{t}{\tau }\right)}^{\beta }\right)=\tau^{1/\beta }\Gamma \left(\left(\beta +1\right)/\beta \right)$$
or
(15)$$\bar{\alpha }=\int_{0}^{\infty }\;dt\;\exp\left(-\alpha t^{\beta }\right)\equiv \alpha^{-1/\beta }\Gamma \left(\left(\beta +1\right)/\beta \right),$$
where here Γ is the gamma function. No source appears to have proposed a physical basis for identifying $\tau_{p}$ or $\bar{\alpha }$ rather than some other average over $\langle X_{pi}\left(t\right)X_{pi}\left(0\right)\rangle$ as an appropriate characteristic time, but this average is simple and broadly used.

The effective relaxation rate for this stretched exponential in time is [20]
(16)$$W^{\mathrm{eff}}=\frac{1}{4\tau_{p}\sin^{2}\left(p\pi /2N\right)}∼\frac{T}{fb^{2}}.$$

For the Rouse model, $W^{\mathrm{eff}}$ only depends on the temperature, the monomer friction factor *f*, and the segment length *b*.

## 3. General Tests of the Rouse Model

This section considers a few general tests of the Rouse model.

First, Rouse assumed that a shear velocity in the *x* direction would only induce bead displacements parallel to the *x* axis. Furthermore, under the influence of a shear field, in the Rouse model, the modes remain uncorrelated, while the relaxation rates are independent of the applied shear. Rouse’s formula for the polymeric viscosity increment is based on these assumptions. In contrast, in the Kirkwood–Riseman model, a shear field causes polymers to rotate. An applied shear field $dv_{x}/dy$ creates bead motions parallel and antiparallel to the *x* axis and also parallel and antiparallel to the *y* axis. Kirkwood and Riseman’s model does not consider the response of polymer internal modes to an applied shear.

To resolve this contradiction between the Rouse and Kirkwood–Riseman models, I made Brownian dynamics simulations on a single bead–spring polymer coil [18]. Hydrodynamic interactions were not included, so the random thermal forces on separate beads could be treated as being uncorrelated. As the test was of the Rouse model, the original Rouse harmonic potential linked all pairs of bonded beads. Chains had no bending constraints. Remote parts of the chain were able to pass ghostlike through each other.

It was shown that the Kirkwood–Riseman model, so far as it goes, is correct, while the Rouse and Zimm models are wrong for a polymer coil in a shear field. In particular, these simulations demonstrated for coils in a shear field that polymer coils do indeed rotate so that
(17)$$\langle \sum_{i=0}^{N-1}y_{i}v_{xi}\rangle =-\langle \sum_{i=0}^{N-1}x_{i}v_{yi}\rangle ,$$
where $x_{i}$ and $y_{i}$ are the *x* and *y* components of bead *i*’s location relative to the polymer center of mass, while $v_{xi}$ and $v_{yi}$ are the *x* and *y* components of bead *i*’s velocity. In addition, in a shear field, the Rouse modes become cross correlated, so the Rouse coordinates cease to represent normal modes. Under shear, the mean-square amplitudes and relaxation rates of the Rouse modes are found to depend on the shear rate. All these results are inconsistent with the Rouse model.

The deficiency in the Rouse model is at its very beginning. Its equations of motion for the polymer beads have no applied shear, so the model only refers to an isolated polymer coil in a quiescent liquid. Rouse polymers therefore do not rotate. However, they are also not subject to a shear field, so they do not create viscous dissipation.

Arising from these theoretical models is the question of what one means by a ’bead’. Conventional polymer molecules are not actually formed from little spheres connected by very thin Hooke’s-law springs. The beads and springs are abstracted from an actual description of a polymer molecule. The Rouse model is not an atomic-level model; rather, atomic motions involving distances shorter than some limiting length have been approximated into the beads and springs. It is conventionally assumed that the limiting length is the Kuhn length, but that assumption is not intrinsic to the Rouse model. In the Kuhn model, a polymer is divided into *N* segments of length *b*, the segments being straight and ‘freely jointed’, i.e., each segment is free to make an arbitrary angle with the next segment in line. *N* and *b* are determined by two constraints, namely that the length of the fully stretched polymer is Nb, while the root-mean-square polymer end-to-end distance *R* satisfies $\langle R^{2}\rangle =Nb^{2}$. The statement that the length is Nb implies that, within each segment, the polymer is fully stretched. On the other hand, in the Kirkwood–Riseman model, the beads are defined to be monomers, the long-range nature of the Oseen hydrodynamic interaction tensor leading to the result that the distribution function for the distances between relatively remote beads is the significant distribution function. In a simulation, no approximations are required; one can actually simulate a Rouse-model chain composed of hydrodynamic beads connected by Hookean springs.

Agapov and Sokolov [21] note that these definitions of *N* and *b* refer purely to polymer statics, but that a dynamic bead of the Rouse model has often been identified with a Kuhn segment. Agapov and Sokolov compare various implicit determinations of bead size with the nominal Kuhn length *b*. The notion of the identification was that the Kuhn length was the length of the shortest chain segment that followed Rouse dynamics, at least in the melt, and that the dynamics of shorter chain segments were not described by the Rouse model. As Agapov and Sokolov explain, for a Rouse chain, the relaxation rate of the intermediate structure factor $g^{\left(1\right)}\left(q,t\right)$, typically as obtained from neutron scattering, scales as $q^{4}$. However, if the scattering vector *q* is made sufficiently large, one probes chain motions over distances less than that for which the Rouse model is valid, in which case at some q=Q, the relaxation rate deviates noticeably from the $q^{4}$ behavior. The corresponding distance 2π/Q defines a dynamic bead size. Agapov and Sokolov note the analysis of Nicholson et al. [22] that in poly(dimethylsiloxane) 2π/Q≈b≈16Å but that in polystyrene 2π/Q≈55Å is about 2.5 times as large as *b*, i.e., the dynamic bead size inferred from $g^{\left(1\right)}\left(q,t\right)$ is perhaps 2.5 times the Kuhn length. Agapov and Sokolov also note computer simulations [23] and oscillatory flow birefringence studies [24] that found a dynamic bead size that is considerably larger than the Kuhn length. They conclude that interpretations of the Rouse model that require that the dynamic bead is the size of a Kuhn length cannot be correct.

Colmenero [15] remarks that the reptation model assumes the validity of the Rouse model at times shorter than the entanglement time $\tau_{e}$. He observes that the two fundamental results of the Rouse model are that the Rouse modes are independent so that $\langle X_{pi}\left(0\right)X_{qj}\left(0\right)\rangle =0$ if p≠q, and that the Rouse correlators $\langle X_{pi}\left(0\right)X_{pi}\left(t\right)\rangle$ decay exponentially in time. Colmenero notes a variety of cases in which the observed correlators relax as stretched exponentials rather than exponentials in time, for example, in cold melts [25] and in blends [26,27,28]. As an interpretation, he suggests that, at low temperatures, non-exponential decay of Rouse correlators might arise from the coupling of polymer motions to local density fluctuations (the α relaxation). Issues then arise from the non-exponential time dependence of the Rouse correlation functions $\langle X_{pi}\left(0\right)X_{pi}\left(t\right)\rangle$. Colmenero emphasizes that various time-dependent physical quantities, such as coherent and incoherent scattering functions and dielectric relaxation spectra, have historically within the Rouse model been calculated incorrectly because the calculations incorrectly assume exponential relaxation of the $\langle X_{pi}\left(0\right)X_{pi}\left(t\right)\rangle$. Therefore, inferential tests of the Rouse model, when based on comparisons of experimental data with Rouse’s formulae for the zero-shear viscosity and other experimental quantities, are invalid because the Rouse correlators do not decay exponentially.

Colmenero [15] proposed to interpret the non-exponential dependence of $\langle X_{pi}\left(0\right)X_{pi}\left(t\right)\rangle$ by replacing the Rouse model’s Langevin equation of motion with a generalized Langevin Equation. Generalized Langevin equations are a natural outcome of the Mori formalism [29,30]; applications to polymer dynamics have been explored by Guenza and colleagues [31,32,33]. Colmenero writes for the time evolution of a Rouse amplitude correlator
(18)$$\frac{d\langle X_{pi}\left(0\right)X_{pi}\left(t\right)\rangle }{dt}+\frac{1}{f}\int_{0}^{t}\;ds\;\Gamma \left(t-s\right)\frac{d\langle X_{pi}\left(0\right)X_{pi}\left(s\right)\rangle }{ds}=-\frac{1}{\tau_{p}}\langle X_{pi}\left(0\right)X_{pi}\left(t\right)\rangle$$
in which Γ(t−s) is a memory function. (An integration by parts would in the integral replace the time derivative of $\langle X_{pi}\left(0\right)X_{pi}\left(s\right)\rangle$ with the function itself). To solve this equation, Γ(t−s) is assumed to be short-lived so that the convolution integral becomes nearly a single-time product, so on defining $\xi \left(t\right)=\int_{0}^{t}ds\Gamma \left(s\right)$, an approximate solution is proposed to be
(19)$$\langle X_{pi}\left(0\right)X_{pi}\left(t\right)\rangle =\langle {\left(X_{pi}\left(0\right)\right)}^{2}\rangle \exp\left(-\frac{f}{\tau_{p}}\int_{0}^{t}ds\frac{1}{f+\xi (s)}\right)$$

On requiring this equation to yield a stretched exponential in *t*, Colmenero proposed that, in the time regime in which ξ(t)≫f, the simplest form for ξ(s) is a power law in *s*. He uses his results to derive a stretched-exponential form for the time autocorrelation function $\langle R_{E}\left(0\right)\cdot R_{E}\left(t\right)\rangle$ of the polymer end-to-end vector $R_{E}$. A comparison was then made with prior atomistic molecular dynamics simulations [27,34] of polyethylene oxide and polymethylmethacrylate/polyethylene oxide melts. $\langle R_{E}\left(0\right)\cdot R_{E}\left(t\right)\rangle$ and the self part of the dynamic structure factor at a series of temperatures were successfully fit to the predicted stretched-exponential time dependencies. Values of $\tau_{p}$ and β at each temperature, as obtained independently from the two physical quantities, were said to be in rather good agreement. Colmenero also compared his results with the Ngai coupling model [35,36].

A rarely tested prediction of the Rouse model states that Rouse modes are orthogonal in the sense that $\langle X_{p}\left(t\right)X_{q}\left(0\right)\rangle =0$ if p≠q. There have been several successful tests of this prediction for the special case t=0, including the results of Kopf et al. [14] and Tsalikis et al. [13]. However, as part of an extended review of theoretical models for viscoelasticity and molecular rheology, Likhtman [16] obtained $\langle X_{1}\left(t\right)X_{3}\left(0\right)\rangle$ for several models of a melt of entangled polymers. While $\langle X_{1}\left(0\right)X_{3}\left(0\right)\rangle \approx 0$ to good approximation, with increasing *t* Likhtman found (Figure 33 of his paper) that $\langle X_{1}\left(t\right)X_{3}\left(0\right)\rangle$ increases substantially with increasing *t*, to far above any noise in the simulation, and then fades away. Likhtman’s result is entirely contrary to expectations from Rouse model dynamics, in which modes are not cross correlated at any time, leading Likhtman to his observation as quoted above that “*This coupling suggests that the Rouse mode description is not very useful for entangled polymers*”.

In this section, we considered the response of a polymer chain to shear, the mapping of Rouse beads onto polymer segments, implications of non-exponential decays of Rouse correlators, and tests of the orthogonality of Rouse modes. None of the outcomes entirely supported the Rouse model or its interpretations.

## 4. Simulations of Rouse Modes

We now turn to simulations that actually examine the properties of Rouse modes.

Dynamic simulations of polymers are readily traced back to the work of Grest and Kremer [10], who simulated a bead–spring model for a polymer chain, in which the beads are subject to independently fluctuating thermal forces, all bead pairs separated by less than a specified distance interact with a Lennard–Jones potential, and each bead’s motions are coupled to a heat bath that supplies a friction term and a thermal driving force. The interaction between bonded beads is represented with a finitely extensible (FENE) potential
(20)$$U_{ij}\left(r_{ij}\right)=-0.5kR_{o}^{2}\ln\left(1-{\left(r_{ij}/R_{o}\right)}^{2}\right),$$
where $r_{ij}$ is the distance between two beads, *k* and $R_{o}$ are simulational parameters, and the potential is set to zero for $r_{ij}>R_{o}$. The above potential is not the harmonic potential used by Rouse, so strictly speaking, this simulation was not a test of the Rouse model. Single chains and rings containing 50–200 beads, and a 200-bead chain with no Lennard–Jones potentials, were examined. The diffusion coefficient, bead mean-square displacement, mean-square center-of-mass displacement, and static structure factor S(q) were calculated. Comparisons were made with theoretical expectations. Chain motions for these single chains were found to be consistent with the Rouse model.

Kremer and Grest [37] then reported their pioneering study of a melt of bead–spring polymers, covering from the short nonentangled (‘Rouse’) regime up to the highly entangled (‘reptation’) regime. In their molecular dynamics simulation, all beads exerted a purely repulsive Lennard–Jones potential and had a FENE bond between next neighbors along each polymer chain, and had a weak frictional force −fv and a corresponding thermal force. Systems with chain lengths *N* from 5 to 400 beads and a total of 250 to 20,000 beads, corresponding to 16 to 100 chains, were studied. The nominal entanglement length was reported to be ≈35 beads, so these simulations included both unentangled and entangled systems. Static properties that were examined include the mean-square end-to-end distance, the radius of gyration, the static structure factor, and the mean-square amplitude of Rouse modes. These quantities show the expected scaling dependencies on *N*. Time-dependencies of the mean-square displacements of single monomers, chain centers-of-mass, and monomers relative to the center of mass, of Rouse amplitudes, of scattering functions, and of chain motion relative to a primitive path were also analyzed. Here, we focus on the Rouse modes.

Kremer and Grest reported the normalized Rouse mode temporal autocorrelation functions
(21)$$\frac{\langle X_{p}\left(t\right)X_{p}\left(0\right)\rangle }{\langle {\left(X_{p}\left(0\right)\right)}^{2}\rangle }$$
for chain lengths N=20,50,100, and 200. Figure 1 shows their results. Kremer and Grest interpreted these as clear single exponentials for the two shorter chains and two time scales for the longer chains. The figures show our fits to stretched exponentials in time. The relaxations are fit well by stretched exponentials. In these pioneering studies, for the two longer chains, at long times, the correlation functions show weak fluctuations on top of the stretched exponentials, making it difficult to determine β accurately. Kremer and Grest also evaluated $\langle X_{p}\left(0\right)X_{q}\left(0\right)\rangle$ for p≠q, finding that the static cross correlations vanish to within the noise in the simulations.

**Figure 1 polymers-15-02615-f001:**
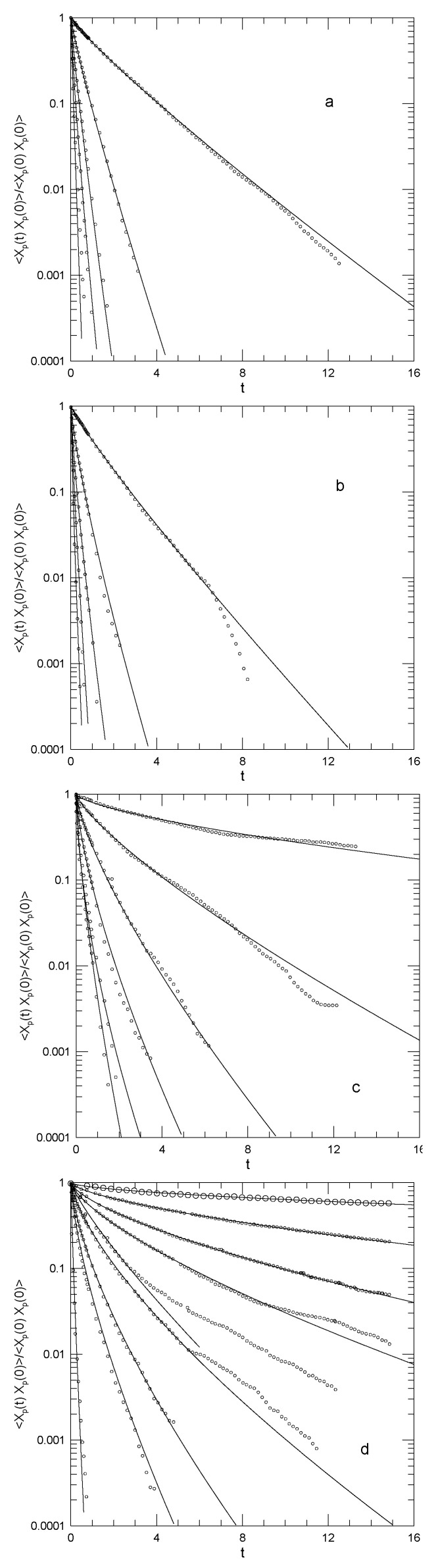
Time autocorrelation functions from Kremer and Grest [37] for Rouse mode amplitudes $X_{p}\left(t\right)$ for p=1,2,3,4,5,6,8,10,20 (top to bottom) for melts of bead–spring linear polymers having *N* of (**a**) 20, (**b**) 50, (**c**) 100, and (**d**) 200. Larger values of *p* do not appear in all graphs; values of *p* are listed in Table 1. Circles represent the original simulations; solid lines are stretched exponentials.

Tsalikis et al. [13] report simulations of ring polymers. Their study is noteworthy for the range of chain parameters that were studied during the course of their simulations. A major focus of the work is comparison with Rouse model predictions for chain dynamics, but a considerable number of other parameters were also studied. These workers report an extended series of molecular dynamics simulations of 5, 10, and 20 kDa poly(ethylene oxide) ring polymers in the melt, corresponding to polymers having 120, 227, or 455 monomers. Simulations were made with a united atom force field [38,39] under isothermal/isobaric conditions, with T=413 K and P=1 atm. The force field parameters were expected to be sufficiently realistic for quantitative comparisons with experiments to be expected to be possible, as confirmed in the paper. For the largest polymer, the simulation cell contained more than 50,000 atoms, the simulation being extended to an equivalent of 2.2 μS.

In considering Tsalikis et al.’s findings on the applicability of the Rouse model to ring melts, one might say that the cup is half full or half empty. Tsalikis et al. chose to emphasize points, where their simulations clearly match the Rouse model predictions. Here, we emphasize the differences, points where the simulations do not match the Rouse model as applied to a ring polymer.

Tsalikis et al. use their simulation data to compute for their rings the mean-square Rouse amplitudes. Their interpretation of a Rouse model for rings predicts that the normalized amplitudes $\langle {\left(X_{p\alpha }\left(0\right)\right)}^{2}\rangle p^{2}/N$ should be independent of mode number *p* and polymer bead count *N*. This prediction is rejected by Tsalikis et al.’s simulations: the normalized amplitudes depend on *p*, and at small *p* are smaller than predicted by the Rouse model. For the N=455 polymer, the normalized amplitude for p=2 is modestly more than half its value for the same polymer at large *p*. The range of smaller *p*-values for which the normalized amplitudes are below their large-*p* limit increases with increasing *N*, the increase in the range being approximately linear in *N*. However, for all *N* studied, the normalized amplitudes do appear to go to the same large-*p* limit, so the model is arguably valid for large *p*.

For each of their chain lengths and p=2,4,6,8,10, and 12, Tsalikis et al. also report the time dependence of the time correlation functions $\langle X_{pi}\left(t\right)X_{pi}\left(0\right)\rangle$. Figure 2 shows a sampling of their measurements (dots). The figure also shows our fits to stretched exponentials (solid lines) and to pure exponentials (dashed lines, obtained from fits to the initial slope). Here, α and β are fitting parameters. The correlation functions are normalized to unity at t=0. If Figure 2 is examined, it is apparent that the relaxation of $\langle X_{pi}\left(t\right)X_{pi}\left(0\right)\rangle$ is described well by a stretched exponential in time, except for a few of the largest-*t* points, contrary to the Rouse model prediction that the relaxations should be simple exponentials.

The stretched exponential is characterized by α and β. Numerical values for the fitting parameters and the average decay rate $\bar{\gamma }$ are seen in Table 2. For each molecular weight, α increases nearly 40-fold between p=2 and p=12. β is close to unity for the N=120 polymer but about 0.8 for the two larger rings.

Tsalikis et al. calculate the normalized cross correlations $\langle X_{pi}\left(t\right)X_{qj}\left(0\right)\rangle$, Equation (Equation 10), between the Rouse amplitudes. They observe that the cross correlations are not large; $\left|\langle X_{pi}\left(t\right)X_{qi}\left(0\right)\rangle \right|$ is almost always less than 0.1. Before considering this result, we ask how large $\langle X_{pi}\left(t\right)X_{qj}\left(0\right)\rangle$ is plausibly likely to be. If $\langle X_{pi}\left(t\right)X_{qj}\left(0\right)\rangle =1$, modes *p* and *q* are perfectly cross correlated; the value of one determines the value of the other. If one mode is cross correlated with several others, $\langle X_{pi}\left(t\right)X_{qj}\left(0\right)\rangle$ for any pair of modes must be considerably less than unity. For example, if a given mode is equally cross correlated with *n* other independent modes, then, at most, the *n* modes completely determine the value of the given mode, in which case the cross correlations would be of typical size 1/n. One might also ask how accurately $\langle X_{pi}\left(t\right)X_{qj}\left(0\right)\rangle$ can be determined. If $\langle X_{pi}\left(t\right)X_{qj}\left(0\right)\rangle$ lies within simulational error of zero, non-zero values for $\langle X_{pi}\left(t\right)X_{qj}\left(0\right)\rangle$ are uninteresting.

Tsalikis et al. [13] evaluated the relaxation of the correlation function
(22)$$C_{uu}\left(t\right)=\langle u\left(t\right)\cdot u\left(0\right)\rangle .$$

For a linear chain, u is the end-to-end vector. Ring polymers have no ends, so u is usefully defined to be a vector from a bead to another bead half-way around the ring. u has two paths to relaxation. First, its magnitude |u| fluctuates around its average value, contributing a relaxation; however, this process cannot relax the correlation function to zero. Second, as the dominant process, $C_{uu}\left(t\right)=\langle u\left(t\right)\cdot u\left(0\right)\rangle$ relaxes by chain reorientation. At long times, u(t) and u(0) cease to be correlated, so their cross-correlation function decays to zero. As shown by the original authors, $C_{uu}\left(t\right)$ follows a stretched exponential in time, with an average relaxation time that increases as $N^{1.9}$, based on the three molecular weights studied. Tsalikis et al. compare the functional form of $C_{uu}\left(t\right)$ that they obtained from their simulations with predictions from the Rouse model. The Rouse predictions for the time dependencies, other than going to zero at a long time, do not resemble the simulation determinations of the time dependence of $C_{uu}\left(t\right)$.

Other quantities studied by Tsalikis et al. [13] include the intermolecular and intramolecular atom–atom radial distribution functions, which have the expected forms. Static structure factors were calculated and found to be in good agreement with experiment. The distributions of end-to-end distances of chain segments of different lengths were calculated as functions of the length of the segments. The distributions were generally not described by Gaussians, especially for the larger rings. In contrast, an initial assumption of the Rouse and Kirkwood–Riseman models is that the end-to-end distances have Gaussian distributions, an assumption leading to the potential of average force between pairs of linked beads. If the bead-to-bead distances do not have Gaussian distributions, the potential of the average force between them is not a Gaussian and does not correspond to a simple spring. Local dynamics were studied using the temporal autocorrelation functions of the torsion angles; the functions were described well with stretched exponentials in time. Finally, these authors asked how many other polymer chains a given chain typically interacts with. As a sensible approximation to this number, they calculated $K_{1}\left(r\right)$, the average number of other chains that had their center of mass within the radius of gyration of the chain of interest. For their ring polymers, $K_{1}\left(r\right)$ was in the range 1.75–2.75. For linear chains having the same three molecular weights, $K_{1}\left(r\right)$ was in the range 8.5–9.5, with $K_{1}\left(r\right)$ increasing as the chain molecular weight was increased from 5 kDa to 20 kDa.

From the above, Tsalikis et al. concluded that the Rouse model is valid for their systems.

Papadopoulos et al. [40] report united atom simulations of polyethylene oxide rings in the melt and in dilute solution in melts of three different linear polyethylene chains. Simulations used the modified TrAPPE force field [38,39] executed with GROMACS [41] held at T = 413 K and 1 atmosphere. Comparison was made with simulated melts of the three linear chains and with experimental studies by Goosen et al. [42] using nuclear spin echo spectroscopy. Goosen et al. concluded that the segmental dynamics of dilute rings in a melt of linear chains are primarily determined by the dynamics of the host polymers. The ring polymers contained 456 monomers, for a molecular weight of 20 kDa, while the linear polymers had molecular weights of 1.8, 10, and 20 kDa, corresponding to chain *N* of 41, 228, and 456. Simulations included 8 rings and 72–720 linear chains with >$10^{5}$ atoms in a simulation cell.

In addition to a series of other static and dynamics properties, Rouse amplitudes $X_{p}\left(t\right)$ were used to compute $\langle {\left(X_{p}\left(0\right)\right)}^{2}\rangle$ and $\langle X_{p}\left(0\right)X_{p}\left(t\right)\rangle$, the former for $N/p^{2}$ from 100 down to <0.02 and the latter for p=2,4,…12. For small *p*, the Rouse model predicts $\langle {\left(X_{p}\left(0\right)\right)}^{2}\rangle ∼N/p^{2}$. For the ring melt and the blends, this result was confirmed for $N/p^{2}>1$. For larger *p*, i.e., $N/p^{2}<1$, $\langle {\left(X_{p}\left(0\right)\right)}^{2}\rangle$ deviates downward from the small-*p* predicted value, attaining at the largest *p* examined perhaps half the value expected from an extrapolation of small-*p* behavior.

Papadopoulos et al.’s determinations of the time correlation functions $\langle X_{p}\left(0\right)X_{p}\left(t\right)\rangle$ appear in Figure 3. They report their determinations as smooth curves, appearing in the figure here as dotted lines. We fit to stretched exponentials (solid lines) and show simple exponentials (dashed lines) where appropriate. There is one behavior for the ring melt and for dilute rings in the 1.8 kDa chain melt (Figure 3a,b), and a somewhat different behavior for dilute rings in melts of the 10 and 20 kDa chains (Figure 3c,d).

Our description of the relaxation functions is not entirely the same as that of Papadopolous et al. Numerical fits, leading to parameters seen in Table 3, clarify issues visible in the figures. In the ring melt, and in dilute solution in the 1.8 kDa linear chains, $\langle X_{p}\left(0\right)X_{p}\left(t\right)\rangle$ shows a stretched-exponential relaxation at earlier times, followed by a sharp transition to a simple-exponential relaxation at later times. The transition, which is especially prominent for p=2 and p=4, occurs at earlier times and smaller values of $\langle X_{p}\left(0\right)X_{p}\left(t\right)\rangle /\langle {\left(X_{p}\left(0\right)\right)}^{2}\rangle$ as *p* is increased. For larger *p*, the transition is more readily apparent in the solution of rings in the 1.8 kDa linear chain melt than in the ring melt. In contrast, for rings in dilute solution in the 10 kDa and 20 kDa melts, $\langle X_{p}\left(0\right)X_{p}\left(t\right)\rangle$ for p=2 and for p=4 relaxes as a single stretched exponential out to the longest times observed. At larger *p*, $\langle X_{p}\left(0\right)X_{p}\left(t\right)\rangle$ fluctuates around the fitted stretched exponential. The transition between short and long time behavior is not seen in many other studies, so it appears to be worthy of further investigation.

Papadopolous et al. report integrated times $\tau_{p}$ for their four systems and the six smallest values of *p*. They report that $\tau_{p}$ scales approximately as ${\left(N/p\right)}^{2}$, $\tau_{p}$ being several-fold larger for rings in melts of the larger-*N* linear polymers than for rings in their own melts. Note, however, that Papadopolous et al. found Gaussian distributions of distances between remote parts of the rings.

Kopf et al. [14] demonstrate a novel simulational test of the Rouse model. They consider systems containing 16 to 25 chains of 10 to 150 beads, in which the forces between all pairs of chains are exactly identical, but in which the beads on some or all of the polymer chains are made 4 or 100 times as massive as the original ’light’ beads. The forces between the beads are the FENE potential and a truncated, purely repulsive bead–bead Lennard–Jones potential. From basic statistical mechanics, a change in bead mass should have no effect on the static properties of the chains in a melt, an outcome that was confirmed simulationally. In mixtures, increasing the mass of the heavier beads slows down the motions of the light-bead chains. Kopf et al. took advantage of the fact that they were doing molecular dynamics to calculate the velocity autocorrelation function through multiple oscillations out to long times. The frequency of the oscillations is relatively independent of the fraction of light or heavy polymers in the system, suggesting that the oscillations in the velocity autocorrelation function arise primarily from intramolecular interactions. The Rouse model remained accurate in light–heavy polymer mixtures. Rouse modes were found not to be cross correlated. Rouse time correlation functions decayed approximately exponentially in time. Contrary to the Rouse model, these simulations observed subdiffusion (mean-square center-of-mass displacement proportional to $t^{0.8}$) on shorter time scales. A nominal entanglement time was used to estimate a nominal tube diameter, which the authors also described as a characteristic length for slowing down the monomer motion. Their results indicated that the nominal tube diameter is independent of the monomer mass, implying that the tube diameter is a static rather than a dynamic quantity, consistent with topological pictures for entanglements.

We turn to Paul et al. [43], who studied 40 to 120 chains of a C_100_ polyethylene using atomistic molecular dynamics. Their simulations included both an explicit-atom model and also a unified atom model, in which each CH_2_ group was treated as a single atom. The polymer was chosen to be long enough that it could reasonably be expected to show Gaussian behavior for its static chain statistics, yet short enough that its dynamics would be expected to have Rouse-like and not reptational behavior. The authors recognized that the assumption of Rouse-like behavior in unentangled melts required examination. A stochastic dynamics simulation was used to equilibrate the samples, while data were obtained using molecular dynamics. Static behavior was tested by calculating the static structure factor; good agreement between simulation and experiment was found. In addition to other dynamic studies, large-scale dynamic behavior was compared with expectations from the Rouse model.

The end-to-end vector reorientation time and the long-time self-diffusion coefficient are consistent with the same value for the segmental friction coefficient, these results being applicable ‘*on time scales larger than the Rouse time*’. However, contrary to the Rouse model, at times shorter than the Rouse time, the center-of-mass diffusion is subdiffusive, being proportional to $t^{0.83}$ or so. Static mean-square amplitudes $\langle {\left(X_{p}\left(0\right)\right)}^{2}\rangle$ of Rouse modes were calculated. For p≤3, the small-*p* Rouse model expectation $\langle {\left(X_{p}\left(0\right)\right)}^{2}\rangle ∼p^{-2}$ was observed. For p>3, the mean-square mode amplitudes decrease approximately as $p^{-3}$, not as $p^{-2}$. The equal-time cross-correlation functions $\langle X_{p}\left(0\right)X_{q}\left(0\right)\rangle$ (p≠q) were found to vanish to “*…within the error bars in the simulation*”; they interpret this finding to imply that “*…the Rouse modes are still good eigenmodes for a dynamic analysis…*”, a claim that would also require that $\langle X_{p}\left(0\right)X_{q}\left(t\right)\rangle \neq 0$ for p≠q and t>0.

Paul et al. also calculated the temporal autocorrelation functions $\langle X_{p}\left(t\right)X_{p}\left(0\right)\rangle$. A plot of correlation functions with p=1,2, and 3 finds that the three correlation functions decay nearly exponentially as $\exp(-\Gamma p^{2}t)$, a single value of Γ sufficing for all three values of *p*, with small deviations over the first quarter of the decay. For p>3, the $\langle X_{p}\left(t\right)X_{p}\left(0\right)\rangle$ are markedly non-exponential. When plotted against $p^{2}t$, with increasing *p*, the $\langle X_{p}\left(t\right)X_{p}\left(0\right)\rangle$ decay more rapidly. The authors conclude that the Rouse model ‘*…is at most applicable to a few largest scale eigenmodes*’. They do, however, note that the self-diffusion coefficient and the rotational diffusion coefficient can be described self-consistently in terms of a single segmental friction factor.

These results were extended by Paul et al. [23], who considered the single-chain intermediate structure factor $g^{\left(1\right)}\left(q,t\right)$ for unentangled polyethylene molecules in a melt, comparing results from neutron spin echo spectroscopy with results from atomistic and from united atom molecular dynamics simulations. They continued to study C_100_ polyethylene, because the polymer is short enough not to be entangled and long enough to have Gaussian chain statistics. The corresponding Rouse model has two parameters, namely a bond strength revealed by the average segment length *b*, and a monomer drag coefficient revealed by the chain center-of-mass diffusion coefficient *D*, the latter determined both experimentally and from each of the two sets of simulations. The simulation values for $g^{\left(1\right)}\left(q,t\right)$ for the explicit-atom and unified-atom simulations are in agreement with the experiments over a factor of six in *q* and two orders of magnitude in the scaled time Dt.

Having validated the accuracy of the simulations, Paul et al. then used the simulations to calculate the Rouse amplitudes, their time autocorrelation functions, and the $g^{\left(1\right)}\left(q,t\right)$ implied by the Rouse modes. The $g^{\left(1\right)}\left(q,t\right)$ predicted by the Rouse model only agrees with the simulations for a limited range of *q* ($\leq 0.14{Å}^{-1}$) and times ≤ 4 nS. At larger *q* and separately at longer times, Rouse-model predictions of $g^{\left(1\right)}\left(q,t\right)$ are significantly smaller than $g^{\left(1\right)}\left(q,t\right)$ from the experiment or as calculated directly in the simulations. The authors note three marked deviations between the simulational results and the Rouse model: first, for times less than a time $\leq \tau_{R}$, diffusion is found by the simulations to be subdiffusive, with exponent 0.83, rather than diffusive; in contrast, Rouse-model chains exhibit diffusive center-of-mass motion at all times. Second, simulations find that only the lowest Rouse modes, p≤3, have relaxations that scale as $p^{2}t$; for larger *p*, Paul et al. attribute deviations from $p^{2}t$ behavior to non-Gaussian chain statistics at short distances. Third, in the simulations, each $\langle X_{p}\left(t\right)X_{p}\left(0\right)\rangle$ decays as a stretched exponential in time; the $\langle X_{p}\left(t\right)X_{p}\left(0\right)\rangle$ of the Rouse model all decay as pure exponentials.

Finally, $g^{\left(1\right)}\left(q,t\right)$ from the simulations, together with a Gaussian approximation
(23)$$g^{\left(1\right)}\left(q,t\right)=\exp\left(-q^{2}\langle {\left(\Delta R_{\mathrm{cm}}\left(t\right)\right)}^{2}\rangle /6\right),$$
was used to calculate a mean-square center-of-mass displacement $\langle {\left(\Delta R_{\mathrm{cm}}\left(t\right)\right)}^{2}\rangle$. It should be emphasized that Doob’s theorem guarantees as a mathematical certainty that if the physical requirements leading to the Gaussian approximation are valid, then, as a mathematical certainty, $\langle {\left(\Delta R_{\mathrm{cm}}\left(t\right)\right)}^{2}\rangle$ increases linearly in time. However, as found by Paul [43], at short times, the calculated center-of-mass motion is subdiffusive, i.e., $\langle {\left(\Delta R_{\mathrm{cm}}\left(t\right)\right)}^{2}\rangle$ grows as $t^{0.83}$ not as $t^{1}$. The Gaussian-approximation estimate of the mean-square displacement does agree with the simulations at long times $t\geq \tau_{R}$, at which the center-of-mass motion is diffusive. At times shorter than the Rouse time, $\langle {\left(\Delta R_{cm}\left(t\right)\right)}^{2}\rangle$ as determined by the simulation is considerably larger than $\langle {\left(\Delta R_{cm}\left(t\right)\right)}^{2}\rangle$ inferred from $g^{\left(1\right)}\left(q,t\right)$ and Equation (Equation 23), showing that the Gaussian approximation is not valid in these systems at shorter times.

Several theoretical advances followed this work. Smith and Paul [44] used quantum chemistry calculations to generate an improved set of force parameters for simulations of 1,4-polybutadiene. Harnau et al. [45,46] proposed that the experimental results of Paul et al. [23] could be understood by replacing the Rouse model with a semiflexible chain model that takes into account chain stiffness. The semiflexible chain model with reasonable parameters agrees well with Paul et al.’s experimental and simulational determinations of $g^{\left(1\right)}\left(q,t\right)$ of the mean-square displacement of the central monomer as a function of time and of the Rouse-mode relaxation times.

Smith et al. [47] present simulations of an unentangled C_100_H_202_ 1,4-polybutadiene melt using a united atom, quantum chemistry-based potential, the focus of the work being to examine the presence of non-Gaussian displacement distributions of polymer beads in a melt. The single-chain intermediate structure factor $g^{\left(1\right)}\left(q,t\right)$ was determined from neutron spin-echo measurements and separately from molecular dynamics simulations using Smith and Paul’s [44] united atom potential. For 0.05≤q≤0.30 Å${}^{-1}$ and times out to 17 nS, experimental and simulated values of $g^{\left(1\right)}\left(q,t\right)$ were in good agreement. The center-of-mass motion was diffusive at long times but subdiffusive $\left(\langle {\left(\delta R\left(t\right)\right)}^{2}\rangle ∼t^{0.8}\right)$ at times shorter than $\tau_{R}\approx 15$ nS. The simulated $g^{\left(1\right)}\left(q,t\right)$ was compared with predictions of the Rouse model and several of this model’s proposed modifications, finding that none of the models reproduced the simulations. Simulations were also used to calculate $\langle X_{p}\left(t\right)X_{q}\left(0\right)\rangle$, the correlation function vanishing for p≠q, at least for p,q≤4. Use of the simulated $\langle X_{p}\left(t\right)X_{p}\left(0\right)\rangle$ in the Rouse form for $g^{\left(1\right)}\left(q,t\right)$ also did not lead to agreement of this modified Rouse model with experiment.

The authors observe that the Gaussian approximation for $g^{\left(1\right)}\left(q,t\right)$ is only appropriate if the distribution of relative bead displacements $R_{m}\left(t\right)-R_{n}\left(0\right)$, $R_{m}\left(t\right)$ being the position of bead *m* at time *t*, is Gaussian at all times. To examine the consequences of this observation, they calculated $g^{\left(1\right)}\left(q,t\right)$ using the Gaussian approximation and values of mean-square displacements $\langle {\left(R_{m}\left(t\right)-R_{n}\left(0\right)\right)}^{2}\rangle$ obtained from their simulations, finding that this calculated $g^{\left(1\right)}\left(q,t\right)$ was in good agreement with $g^{\left(1\right)}\left(q,t\right)$ as predicted by the Rouse model but did not agree with $g^{\left(1\right)}\left(q,t\right)$ as calculated directly from the simulation, thus showing the importance of non-Gaussian particle displacements. Smith et al. conclude that the non-Gaussian distribution of bead displacements $R_{m}\left(t\right)-R_{n}\left(0\right)$ is responsible for the observed failure of the Rouse model in polymer melts.

Two sorts of non-Gaussian behavior are possible here. The first is that the distribution of displacements $P(\Delta R_{m}\left(\Delta t\right))$ for each bead separately could be non-Gaussian. The second is that the distributions of displacements of pairs of beads could be cross correlated. Thanks to the fluctuation–dissipation theorem, this latter possibility is equivalent to the statement that there are significant hydrodynamic interactions in polymer melts, a possibility that would only be surprising if the Rouse model were correct in polymer melts.

Harmandaris et al. [48] made atomistic simulations of 24-, 78-, and 156-atom (mean length) linear polyethylene melts, finding a diffusion coefficient *D* as well as the time autocorrelation functions of the polymer end-to-end vector and the Rouse mode amplitudes. Each autocorrelation function may be said to have a characteristic time τ. The study was novel, in that the authors deliberately simulated polydisperse melts, having a polydispersity index near 1.09. From these quantities, nominal monomer friction factors were extracted. Initial chain configurations were equilibrated using the end-bridging Monte Carlo scheme [49]. Molecular dynamics were executed using a sixth-order predictor–corrector model. The objectives of the study were to test the Rouse model and to take advantage of the polydispersity to examine the dynamics of chains having different lengths, all in the same melt. Potential energies included a Lennard–Jones potential between non-bonded atoms, bond-bending and torsional potentials, and a Fixman potential [50] to keep the bond lengths constant. The 24- and 78-atom carbon models were simulated in both the NVE and NVT ensembles; the results agreed. The mean-square end-to-end distance $\langle {\left(R\left(0\right)\right)}^{2}\rangle$, radius of gyration, and intermolecular bead–bead static correlation functions from the molecular dynamics simulation and the end-bridging Monte Carlo simulation were found to agree, confirming the validity of the two simulations. Local dynamics as estimated with the torsion angle temporal autocorrelation function showed that local dynamics become slightly slower as the chain length is increased.

Harmandaris et al. [48] calculated properties of the Rouse amplitudes $X_{p}\left(t\right)$. The mean-square static amplitudes $\langle {\left(X_{p}\left(0\right)\right)}^{2}\rangle$ decrease with the increasing *p*, for the first two modes with the $p^{-2}$ dependence predicted by the Rouse model, and at larger *p* much more rapidly, approximately as $p^{-3}$. For p=10, the discrepancy between the simulation and the Rouse model approaches an order of magnitude. The temporal correlation functions $\langle X_{p}\left(0\right)X_{p}\left(t\right)\rangle$, at least for the 83- and 117-carbon chain systems, also do not agree with the Rouse model, namely they are not simple exponentials, and their relaxation times do not scale with time as $p^{2}t$. On the other hand, for the end-to-end vector, the relaxation time of 〈r(0)·r(t)〉 calculated using the Rouse model and the relaxation time inferred from simulations of $\langle X_{1}\left(0\right)X_{1}\left(t\right)\rangle$ agree well. From observations of the chain center-of-mass motion over long times, a chain diffusion coefficient *D* and therefore a monomer friction factor ζ can be inferred. Contrary to the Rouse model, ζ depends on the chain length, increasing threefold from the shortest to the longest chains studied. However, ζ appears to reach an asymptotic value for the longer chains, those with N>60 or so. Harmandaris et al. propose that N=60 is therefore a lower limit, below which the Rouse model is inapplicable. These workers also calculated, from the diffusion coefficient, a zero-shear viscosity. The calculation was based on Rouse’s theory, which, in some other respects, does not describe the dynamics of these systems, but the calculated viscosity determined by $\tau_{1}$ and the p=1 mode does approximately follow Rouse model predictions.

Krushev et al. [51] simulated melts of 1,4-polybutadiene. Their interest was to determine the effects of torsion barriers on molecular motions. To do this, they examined the static structure factor, Rouse mode amplitudes, Rouse–Rouse temporal autocorrelation functions, and the intermediate scattering function $g^{\left(1\right)}\left(q,t\right)$. Their polymer melts incorporated 40 polymer chains, each with 29 or 30 subunits, all with united atom potentials, including a model with chains incorporating vinyl groups, a model with chains not incorporating vinyl groups, and a model with no vinyl groups and all rotational potentials set to zero. The three models have the same distribution for their radii of gyration.

Rouse mode amplitudes had, at most, weak cross correlations, $\langle X_{p}\left(0\right)X_{q}\left(0\right)\rangle$ for p≠q being less than 2% of $\langle {\left(X_{p}\left(0\right)\right)}^{2}\rangle$. The mode amplitudes $\langle {\left(X_{p}\left(0\right)\right)}^{2}\rangle$ only followed the Rouse prediction $\langle {\left(X_{p}\left(0\right)\right)}^{2}\rangle ∼p^{-2}$ for p<5. At larger *p*, the mean-square amplitude depended on good accuracy as $p^{-3}$, which is the prediction for a freely rotating polymer [52]. The calculated static structure factor was not affected by adding or deleting the torsion potential. The time correlation functions $\langle X_{p}\left(t\right)X_{p}\left(0\right)\rangle$ showed stretched-exponential, not single-exponential, behavior for all *p* that were studied over an adequate time range. Time correlation functions with and without torsion potentials were very nearly the same when plotted in reduced time units, in which the 1/e time was identified as unit time, showing that chain stiffness arising from torsion potentials did not have a significant effect on the form of the relaxation of $X_{p}\left(t\right)$.

The intermediate scattering function $g^{\left(1\right)}\left(q,t\right)$ also did not follow the Rouse model predictions, namely the Rouse model predicts an over-rapid decay of $g^{\left(1\right)}\left(q,t\right)$ at larger *q*, and underpredicts the degree of stretching of $g^{\left(1\right)}\left(q,t\right)$. $g^{\left(1\right)}\left(q,t\right)$ is not significantly changed when torsional potentials are added or removed from the potential energy; the authors infer that observed deviations from Rouse behavior are not due to internal rotation barriers. The Rouse model calculation of $g^{\left(1\right)}\left(q,t\right)$ agrees with the $g^{\left(1\right)}\left(q,t\right)$ calculated from the simulation on using the Gaussian approximation and the measured mean-square displacement. Neither calculation agrees with the actual $g^{\left(1\right)}\left(q,t\right)$. Furthermore, the center of mass mean-square displacements are subdiffusive at short times. *The authors conclude that the Rouse model assumption—that atomic motions are described by a joint Gaussian random process—is thus shown to be incorrect.*

Padding and Briels [53] reported simulations of twelve chains of a C_120_H_242_ polyethylene melt. They simulated four different starting points for their melt, molecules having united atom potential energies. The potential energy used by these authors was not the simple-harmonic bond-length potential of the Rouse model. Bond lengths and angles had harmonic potential energies; a torsion potential energy was present; unbonded (separated by four or more atoms in a single molecule) pairs of united atoms had a Lennard–Jones potential. A weak coupling to a bath held the temperature fixed. Melt starting chain configurations were created by gradual compression of a dilute system, in which only repulsive interatomic forces were present.

The time-dependent dynamic parameters that they obtained from their simulations include mean-square displacements, the end-to-end vector time autocorrelation function, the intermediate structure factor, and the stress tensor. A single set of numerical parameters for the number of segments *N* in a chain, for the diffusion coefficient *D*, and for the longest relaxation time $\tau_{1}$, described most of the dynamic quantities that they calculated, but only over distances longer than a limiting length scale. This paper actually tested the relationships between *N*, *D*, $\tau_{1}$, and the calculated dynamic parameters, as predicted by the Rouse model, but did not test the Rouse model itself (Rouse’s description of the internal dynamics of a polymer chain).

Here, the stress tensor was calculated as
(24)$$\sigma =\frac{1}{V}\sum_{i=1}^{N}m_{i}v_{i}v_{i}+\sum_{i\geq j=1}^{N}\left(r_{i}-r_{j}\right)F_{ij}$$
where $v_{i}$ is the velocity of atom (or center of mass) *i*, $r_{i}$ is the position of atom (or center of mass) *i*, and $F_{ij}$ is the force exerted on atom (molecular center of mass) *i* by atom (molecular center of mass) *j*. The forces between two molecules can create a torque, an antisymmetric part of the stress tensor, on each molecule. The stress tensor was identified as leading to the zero-shear relaxation modulus via
(25)$$G\left(t\right)=\frac{V}{10k_{B}T}\langle P\left(t\right):P\left(0\right)\rangle ,$$
where P is the symmetrized traceless part of σ.

Padding and Briels concluded that there is a shortest length scale b≈1.2 nm over which the Rouse model is valid in their simulations. The length scale manifests itself as the shortest Rouse-mode wavelength for which the model works, the shortest distance over which a mean-square displacement must occur before Rouse behavior is seen, and the shortest wavelength for which the simulated $g^{\left(1\right)}\left(q,t\right)$ agrees with the model. Over shorter distances and times, matters become more complicated. Padding and Briels calculate $g^{\left(1\right)}\left(q,t\right)$ for a series of wave vector magnitudes. At small *q*, only center-of mass diffusion is seen. At larger *q*, $g^{\left(1\right)}\left(q,t\right)$ has contributions from polymer internal modes. At larger *q*, $g^{\left(1\right)}\left(q,t\right)$ from the simulation, and $g^{\left(1\right)}\left(q,t\right)$ calculated from the Rouse modes and the diffusion coefficient found at small *q*, do not agree. $g^{\left(1\right)}\left(q,t\right)$ from the Rouse formula decays faster at long times than $g^{\left(1\right)}\left(q,t\right)$ from the simulations, the discrepancy becoming larger at larger *q*. The short-distance internal chain modes are thus not the same as the internal modes predicted from the Rouse model. Similarly, measurements of mean-square displacements from simulations only agree with the Rouse model when mean-square displacements are greater than (1.1 nm)${}^{2}$. Finally, at short times, the simulated shear relaxation modulus G(t) “*…does not behave Rouse-like at all…*”, but corrections due to this issue at long times are limited in size.

In a further paper, Padding and Briels [54] reported simulations of a heavily coarse-grained (one bead = 20 monomers, 120 chains in a simulation box) C_120_ linear polyethylene that incorporated a complicated switchable scheme for enforcing chain uncrossability. The scheme could be turned off, leading to simulations of a melt, in which polymer chains could pass through each other. Interactions included non-bonded, bonded, and bending-angle contributions to the bead–bead potential of average force. Beads incorporated as many monomers as feasible without making the beads larger than a nominal tube radius. Padding and Briels calculated the mean-square displacements, both of single blobs and of the chain centers of mass. For the unentangled chains, mean-square displacements increased linearly with time. Adding the uncrossability constraint reduced the mean-square displacements and gave them a sublinear time dependence over a considerable time regime.

Padding and Briels [54] calculated the time-dependent Rouse amplitudes and evaluated their time correlation functions. Fits were then made to stretched exponentials in time. Chains had six beads, so they only had five internal Rouse modes. When chain crossing was permitted, the time correlation functions were very nearly single exponentials. Adding chain crossing constraints and a bond bending potential led to appreciably non-exponential relaxations, the stretching parameter β falling from close to unity in the absence of chain crossing constraints or a bond bending potential to 0.77 for the three highest modes when the constraint and potential were added. The chain crossing constraint considerably increased the relaxation times of the p=1 and p=2 modes but did not increase substantially the relaxation times of the three higher-*p* modes. Padding and Briels determined effective relaxation times $\tau_{p}^{\mathrm{eff}}$ for their five modes. However, instead of calculating $\tau_{p}$ from the stretched-exponential fitting parameters, the authors performed numerical integrals of the measured $\langle X_{p}\left(t\right)X_{p}\left(0\right)\rangle$ curves. The Rouse relaxation rates, Equation (Equation 16), were evaluated for each *p*. In the Rouse model, $W_{p}^{\mathrm{eff}}$ is independent of *p*. In the presence of chain stiffness, and more dramatically in the presence of uncrossability, $W_{p}^{\mathrm{eff}}$ was found to increase several-fold as *p* was increased, the major change from the Rouse model being that, in the presence of chain stiffness and uncrossability, $W_{p}^{\mathrm{eff}}$ is reduced for small *p*. Finally, Padding and Briels [54] calculated the system’s intermediate structure factor $g^{\left(1\right)}\left(q,t\right)$ for a series of values of *q*. At small *q*, $g^{\left(1\right)}\left(q,t\right)$ is relaxed by whole-body translational diffusion, a fit giving the polymer’s diffusion coefficient. In the presence of chain uncrossability and larger *q*, $g^{\left(1\right)}\left(q,t\right)$ did not agree with the Rouse model predictions.

Padding and Briels [20] further extended their work on polyethylene by making extended united atom simulations of melts of seven different polyethylenes, using 80 to 180 chains with 80 to 1000 carbon atoms coarse-grained into 4 to 50 blobs at 450 K and a density 0.761 g/cm^3^. They stressed that the eliminated internal coordinates become thermal bath variables, and contributed to the motion of the blobs as unseen random thermal forces and a friction factor, which they treated as a scalar with no associated memory function. The paper considers a considerable list of different dynamic parameters. Here, we are only concerned with the behavior of the Rouse amplitudes. They calculated the Rouse amplitudes and their time correlation functions. On plotting $8N\sin^{2}\left(p\pi /2N\right)\langle {\left(X_{p}\left(0\right)\right)}^{2}\rangle$ against N/p, values of $8N\sin^{2}\left(p\pi /2N\right)\langle X_{p}{\left(0\right)}^{2}\rangle$ for all chain lengths *N* superpose, but contrary to the Rouse model $8N\sin^{2}\left(p\pi /2N\right)\langle X_{p}{\left(0\right)}^{2}\rangle$ is not independent of N/p; it instead falls off from slightly more than 3 to slightly more than 1 as N/p is reduced, i.e., as *p* is increased. They interpret this non-Rouseian behavior as arising from their angular potential, which leads to some degree of chain stiffness.

Padding and Briels [20] also calculated the Rouse–Rouse time correlation functions. Rouse modes at time zero are not cross correlated; they did not report what happens at later times. Rouse-mode temporal autocorrelation functions were found to decay as stretched exponentials in time. The stretching parameter β was close to 0.7 near N/p=1. β decreased to 0.55 or so near N/p=2, and then increased to near 0.7 at N/p=6. At larger N/p, β was nearly constant. The dependence of $W_{p}^{\mathrm{eff}}$ on N/p was examined. For N/p≈1, $W_{p}^{\mathrm{eff}}$ is independent of N/p. For N/p, modestly above 1 and up to 3 or so, $W_{p}^{\mathrm{eff}}∼{\left(N/p\right)}^{-2}$. For N/p>3, $W_{p}^{\mathrm{eff}}∼{\left(N/p\right)}^{-1}$. Padding and Briels note, for the second and third regimes, that these dependencies are not in agreement with either the Rouse or the reptation model. Padding and Briels then propose that at large times $\langle X_{p}\left(t\right)X_{p}\left(0\right)\rangle$ switches over from a stretched-exponential to a simple-exponential time dependence.

Abrams and Kremer [55] studied a bead and spring polymer melt, the interest being the effects of varying the equilibrium bond length $\ell_{0}$ relative to a nominal bead diameter $d_{0}$. In different simulations, the bead length was given 13 values in the range $0.73\leq \ell_{0}/d_{0}\leq 1.34$. The model contained 80 freely jointed chains, each having 50 beads, at density 0.85 $\sigma^{-3}$ and nominal temperature T=1 in natural units, so that $d_{0}=\sigma$. Bonded beads were linked with a harmonic potential $\frac{k}{2}{\left(\ell -\ell_{0}\right)}^{2}$, *ℓ* being a bond length, with *k* and $\ell_{0}$ being simulational parameters. Non-bonded beads interacted with a truncated Lennard–Jones potential. The authors studied the time correlation functions of the Rouse amplitudes $X_{p}\left(t\right)$ and the mean-square displacements of the bead centers of mass.

Semi-log plots of the normalized $\langle X_{p}\left(t\right)X_{p}\left(0\right)\rangle$ as functions of a normalized time $tp^{2}/N^{2}$ were presented for $\ell_{0}/d_{0}$ equaling 0.79, 0.97, and 1.24 and p≤5. For $\ell_{0}/d_{0}=0.79$, the plots were nearly linear. The curvature increased with increasing $\ell_{0}/d_{0}$. For the smallest $\ell_{0}/d_{0}$, plots of $\langle X_{p}\left(t\right)X_{p}\left(0\right)\rangle$ for the different values of *p* nearly superpose. For larger $\ell_{0}/d_{0}$, the curves spread out modestly from each other, though the dependence on *p* is hard to discern. Abrams and Kremer extracted $\tau_{p}$ from fits of $\exp(-t/\tau_{p})$ to the early parts of the $\langle X_{p}\left(t\right)X_{p}\left(0\right)\rangle$ curves, and advanced from there to nominal friction factors $\zeta_{p}$. In doing this, they indicated that “*…The Rouse model best describes the early phase of the decay of*$\langle X_{p}\left(t\right)X_{p}\left(0\right)\rangle$…” and described behavior at later times as “*…subtle non-Rouse-like behavior …*”, whose examination was deferred to a future study. The inferred $\zeta_{p}$ values were presented as averages over *p*. As a function of $\ell_{0}/d_{0}$, the averaged $\zeta_{p}$ increases rapidly at larger $\ell_{0}/d_{0}$. Abrams and Kremer also calculated the average number of other polymer beads within a distance $2^{1/6}\sigma$ of a bead of interest. That number increases, roughly from 0.4 to 1, over the observed range of $\ell_{0}/d_{0}$.

Doxastakis et al. [56] reported extensive atomistic and unified-atom simulations of melts of very short (40–115 atom) polyisoprenes, including 64 chains of C_80_ polyisoprene, and compared with measurements from ^13^C NMR, quasielastic neutron scattering, the torsional correlation function from the simulation, dielectric relaxation spectroscopy, and polymer self-diffusion. Because these authors performed atomistic simulations, their simulations determined single-bond and few-atom motions that could be compared with ^13^C NMR and neutron scattering. Reasonable agreement between simulation and experimentally measured quantities, within the expected limits of accuracy of the simulations, was obtained. Dielectric relaxation measurements were interpreted in terms of a Kohlrausch–Williams–Watts function for a higher-frequency peak and Rouse normal modes for a lower-frequency peak. The Rouse fit showed some deviation from the experiment at higher frequencies. The mean-square amplitudes $\langle {\left(X_{p}\left(0\right)\right)}^{2}\rangle$ of the Rouse modes only followed the theoretical $p^{-2}$ scaling for the first two or three modes; for larger *p*, the measured amplitudes are smaller than the theoretical prediction. Plots of the simulated $\langle X_{p}\left(0\right)X_{p}\left(t\right)\rangle$ against $p^{2}t$ should collapse onto a single line. If the t=0 amplitude is normalized out, the plots come respectably close to doing so. However, at short times, $\langle X_{p}\left(0\right)X_{p}\left(t\right)\rangle$ from the simulations fell well below a fit of the long-time $\langle X_{p}\left(0\right)X_{p}\left(t\right)\rangle$ to a single exponential, especially for larger *p*. The simulated time autocorrelation function for the chain end-to-end vector was at early times also smaller than expected from the Rouse model. Finally, on uniting the various theoretical and fitted treatments of chain end-to-end relaxation, very good agreement was obtained between the theoretical form and the simulations. The authors concluded that the Rouse model is sustained by their simulations for the quantities that they analyzed, a conclusion that neglects the issues they faithfully reported with the mode mean-square amplitudes.

Tsolou et al. [11,57,58] report a series of molecular dynamics simulations of polybutadiene and polyethylene. Their first paper [57] simulated *cis*-1,4-polybutadiene based on a united atom description, in which hydrogen atoms were merged with the carbon atom to which they were bonded. Bonds were represented as Hookian springs with a finite rest length; bend and torsion angles had associated potential energies. Non-bonded atoms interacted with a non-truncated Lennard–Jones potential. Melt simulations were performed on monodisperse polymers with *N* of 32 to 400 carbon atoms in systems containing more than 10,000 united atoms for times up to 600 nS. End-bridging Monte Carlo methods were used to create rapid equilibration; simulations were based on multiple-time-step molecular dynamics. The system was thermostatted to a constant temperature and pressure. A long series of static quantities were calculated, including the mean-square radius of gyration, mean-square end-to-end distance, characteristic ratio, specific volume as a function of chain length, density as a function of temperature, the intermolecular pair radial distribution function, and the static structure factor. For the last of these, the locations of the hydrogen atoms had to be backed out from the unified atom description.

Tsolou et al. [57] also calculated dynamic quantities, including the time autocorrelation functions for the torsion angles and the chain end-to-end vector $R_{E}\left(t\right)$. Efforts to fit $\langle R_{E}\left(t\right)\cdot R_{E}\left(0\right)\rangle$ as a sum of Rouse modes were unsatisfactory; on the other hand, $\langle R_{E}\left(t\right)\cdot R_{E}\left(0\right)\rangle$ was fit accurately with a single stretched exponential in *t*. The nominal relaxation times from these fits increased with increasing polymer length *N*, namely $\tau ∼N^{a}$ with *a* increasing from 2.1 for the shortest polymers to 2.8 for the longest polymers. The change in *a* with increasing *N* was not obviously discontinuous.

Tsolou et al. also determined the mean-square displacements of the chain centers-of-mass as functions of time, and inferred from the displacements the diffusion coefficient *D*. *D* depends on *N* approximately as a power law. Curiously, the slope of log(D) against log(N) is shallowest for intermediate values of *N*.

An algorithm was used to obtain a nominal primitive path for each polymer chain at a series of times. The primitive path from this algorithm is a smooth curve that follows the atomistic backbone. The diameter *a* of the corresponding tube is 64Å, which is considerably larger than the experimental 38Å tube diameter reported [59] for the same system. *a* is much larger than the distance between neighboring polymer chains, showing that when a polymer chain attempts to move transversely to its primitive path and encounters another polymer chain, in general, it is able to continue to move in the same direction over considerable distances. The authors also computed the mean-square displacements g(t) of the central beads of each chain. g(t) appears to be a smooth curve that could be described as having sections that follow power laws $t^{\alpha }$. However, α was never less than 0.4, so it never reached the α=0.25 of the Rouse model. For N=400, the transition from an initial α=0.5 down to α=0.4 occurred at ≈$10^{3}$ pS.

Calculations of the single-chain intermediate structure factor $g^{\left(1\right)}\left(q,t\right)$ were made. The authors concluded that there is no *q* for which $g^{\left(1\right)}\left(q,t\right)$ agrees with a fit to the Rouse model, including times shorter than a few nS. According to the authors, the tube-reptation model indicates that at times shorter than a few nS, polymer chains are supposed to be performing Rouse-like motion because they have not yet having encountered the walls of their tube. However, $g^{\left(1\right)}\left(q,t\right)$ obtained from these simulations decays more slowly than does $g^{\left(1\right)}\left(q,t\right)$ predicted by the Rouse model, using Rouse times calculated by fitting independently to the time correlation functions of the polymer end-to-end vector. The tube-reptation prediction does not match the simulations. Nonetheless, the authors were able to extract a nominal friction factor ζ from the Rouse form for the diffusion coefficient, even though the Rouse model does not appear to describe the dynamics.

Tsolou et al. [58] examined Rouse amplitudes and dynamic structure factors for simulated cis-1,4-polybutadiene melts. The simulations viewed 32 chains of a C_128_ polymer as functions of the system’s temperature and pressure, at temperatures from 165 to 413 K and pressures from one atmosphere to 3.5 kbar. Simulation methods duplicated those in earlier papers by Tsolou et al. [60,61]. The authors first examined the time autocorrelation functions $\langle X_{p}\left(0\right)X_{p}\left(t\right)\rangle$ of the Rouse mode amplitudes for *p* having various values in the range (1,64). The $\langle X_{p}\left(0\right)X_{p}\left(t\right)\rangle$ were all found to decay as stretched exponentials in time, not the exponentials predicted by the Rouse model, leading to a set of values for $\tau_{p}$ and $\beta_{p}$. The stretched exponentials were also characterized via their total correlation times
(26)$$t_{p}=\frac{\Gamma (1/\beta_{p})}{\beta_{p}}\tau_{p}.$$

Tsolou et al. [58] found that the $t_{p}$ depends on temperature via a modified Vogel–Fulcher–Tamman equation,
(27)$$t_{p}\left(T\right)=t_{p0}\exp\left(\frac{D_{p}T}{T-T_{o}}\right).$$

Here, *T* is the absolute temperature, $t_{p0}$ and $D_{p}$ are fitting parameters, and $T_{o}$ is a characteristic temperature. Over the range of temperatures that were examined in the simulations, the temperature dependencies of the $t_{p}$ do not depend markedly on *p*. Pressure dependencies of the $t_{p}$ were obtained at temperatures 310 and 413 K. The $t_{p}$ increases exponentially with increasing pressure *P*. Viewed graphically, the effect of *P* on $t_{p}$ does not depend a great deal on *p*. The authors define an activation volume for the $t_{p}$ via
(28)$$\Delta V_{p}\left(P\right)=RT{\left(\frac{\partial \ln(t_{p})}{\partial P}\right)}_{T}.$$
$\Delta V_{p}$ decreases roughly by two-fold between p=1 and p=64, and for smaller *p* is modestly smaller (by less than 10%) at the higher than at the lower temperature.

Tsolou et al. [58] also calculated the single-chain intermediate structure factor [57,62]
(29)$$g^{\left(1\right)}\left(q,t\right)=\sum_{n,m=1}^{N}\sin\left(qR_{nm}\left(t\right)\right)/qR_{nm}\left(t\right)$$
in which *n* and *m* label two of the *N* segments of a single chain, *q* being the magnitude of the scattering vector and $R_{nm}\left(t\right)$ being the distance between segments *n* and *m* at two times separated by *t*. $g^{\left(1\right)}\left(q,t\right)$ was found to be described by a stretched exponential in *t*. The stretching exponent β was reported to change with pressure and to decrease with increasing *q*. The total correlation times for $g^{\left(1\right)}\left(q,t\right)$ were found to decrease with increasing *q* and to increase exponentially with increasing pressure. The activation volumes ΔV(T), as calculated from the pressure dependencies of these total correlation times, decrease with increasing *q*.

Tsolou et al. [58] calculated the single-chain incoherent scattering factor $g^{\left(1s\right)}\left(q,t\right)$, which differs from $g^{\left(1\right)}\left(q,t\right)$ in that in Equation (Equation 29) the restriction n=m is forced. $g^{\left(1s\right)}\left(q,t\right)$ decays as a stretched exponential in time. β was found to increase from ≈0.5 to ≈0.9 as *q* was increased over the observed range. The total correlation time $t_{c}$ decreased strongly with increasing *q*. Lines to guide the eye, drawn as $\tau ∼q^{-2/\beta }$ for smaller *q* and $\tau ∼q^{-2}$ at larger *q*, are harmonious with the *q* dependencies of $t_{c}$ at multiple temperatures and a full range of pressures. The peculiar −2/β exponent is an artifact of the stretched-exponential form $\exp(-{\left(t/\tau \right)}^{\beta })$ used to parameterize $g^{\left(1s\right)}\left(q,t\right)$. If the parameterization had instead been the rational alternative
(30)$$g^{\left(1\right)}\left(q,t\right)=g^{\left(1\right)}\left(q,0\right)\exp\left(-\Gamma t^{\beta }\right),$$
then for smaller *q*, the result would have been $\Gamma ∼q^{2}$, while for larger *q*, the form $\Gamma ∼q^{2\beta }$ would have appeared as an approximant.

The Rouse model predicts that the relaxation time of the $p^{\mathrm{th}}$ mode should depend on *p* as $p^{-2}$. As the model also predicts that modes relax as simple exponentials in *t*, not the stretched exponentials actually found, there is no theoretical basis for identifying the total correlation time with the Rouse time. Indeed, $p^{2}t_{p}$ is not independent of *p*; it instead decreases by about 30% as *p* is increased from 1 to 20. Tsolou et al. [58] used this observation to estimate the longest relaxation time and hence a model-dependent zero-shear viscosity for the system. The inferred viscosity follows a Vogel–Fulcher–Tamman form.

Tsolou et al. [11] simulated melts of ring polyethylenes containing 24–400 carbon atoms per ring at nominal temperature and pressure of 450 K and 1 atmosphere; simulation boxes contained 3000–20,000 united atoms. Simulations were made with a united atom model treating methylene units as single atoms, a harmonic bond-stretching potential, a harmonic-in-bond-angle bending potential, a bond-torsional potential, and a 12-6 Lennard–Jones potential for atoms separated by more than three bonds, using the r-RESPA algorithm [63] for molecular dynamics. As two of a large number of properties (most not considered in this review), they calculated $\langle {\left(X_{p}\left(0\right)\right)}^{2}\rangle$ and $\langle X_{p}\left(0\right)X_{p}\left(t\right)\rangle$. For $N/p^{2}\geq 2$, the mean-square amplitude $\langle {\left(X_{p}\left(0\right)\right)}^{2}\rangle$ increased linearly in $N/p^{2}$. At smaller $N/p^{2}$, the increase in $\langle {\left(X_{p}\left(0\right)\right)}^{2}\rangle$ with increasing $N/p^{2}$ was much more rapid than linear in $N/p^{2}$, contrary to any prediction of linear behavior at all $N/p^{2}$. The time correlation functions $\langle X_{p}\left(0\right)X_{p}\left(t\right)\rangle$ were found to depend on *t* as stretched exponentials in time, not the simple exponentials of the Rouse model. A comparison was made between a Rouse prediction $\tau ∼N^{2}/p^{2}$ for small *p*, and the $\tau_{p}$ calculated from the time integral of $\langle X_{p}\left(0\right)X_{p}\left(t\right)\rangle$. The prediction was sustained for N/p≥30. At smaller N/p, $\tau_{p}$ decreases two-fold as N/p is reduced from 30 toward 1. Rouse model predictions for the time correlation function and its zero-time value are, therefore, confirmed only for larger values of N/p.

Bulacu and van der Giessen [64] simulated the effect of bending and torsional potential energies on a polymer melt. Their systems included up to 1000 chains with 5≤N≤250. Their polymer was a bead–spring model, with a 6-12 Lennard–Jones potential truncated at its minimum, bonds between beads represented with a FENE potential, a cosine harmonic bending potential
(31)$$V\left(\theta \right)=\frac{1}{2}k_{\theta }{\left(\cos\left(\theta \right)-\cos\left(\theta_{0}\right)\right)}^{2}$$
with $\theta_{0}=109.5^{o}$, and a coupled bending–torsion potential
(32)$$V=k_{\phi }\sin^{3}\left(\theta_{i-1}\right)\sin^{3}\left(\theta_{i}\right)\sum_{n=0}^{3}a_{n}\cos^{n}\left(\phi_{i}\right).$$

The torsion potential refers to four beads in a row along a polymer, the two internal angles $\theta_{i-1}$ and $\theta_{i}$ formed by the two overlapping sets of three beads in a row, and the dihedral angle ϕ. The $a_{n}$ was determined by quantum calculations for *n*-butane as $\left(a_{0},a_{1},a_{2},a_{3}\right)=\left(3,-5.9,2.06,10.95\right)$. Simulations used the velocity Verlet algorithm; the temperature was held steady with a heat bath’s random force and friction factor. $k_{\theta }$ and $k_{\phi }$ are stiffness coefficients.

Static properties examined include the mean-square end-to-end distance $\langle R^{2}\rangle$ and the radius of gyration $R_{g}$; these were calculated both for the entire chain and also for all of its sub-chains. With increasing chain stiffness, $\langle R^{2}\rangle /\langle R_{g}^{2}\rangle >6$ was found. Histogram distributions of bond lengths, bending angles, and dihedral angles were reported. The bead–bead radial distribution functions, same chain, different chain, and all chains, were reported as linear plots. (Linear plots, while completely orthodox, can lose details of g(r). Whitford and Phillies previously showed that the range of g(r) in a Lennard–Jones fluid is, at lower temperatures, much longer than is sometimes assumed [65,66] as was made apparent by plotting log(|g(r)−1|) against *r*).

Dynamic properties studied include the polymer self-diffusion coefficient *D*, mean-square bead displacements, and $\langle X_{p}\left(t\right)X_{p}\left(0\right)\rangle$. For the N=40 chain, *D* depends on the stiffness coefficients as $k_{\theta }^{-0.48}$ and $k_{\phi }^{-0.52}$. *D* depends on *N* via a smaller-*N* and a larger-*N* power law. The authors determined the break between the two power laws by maximizing the sum of the regression coefficients in fits of the two laws to the data. With increasing chain stiffness, *D* is reduced. With increasing chain stiffness, for shorter chains, *D* depends more strongly on *N*, the transition from short-to long-chain behavior moves to larger *N*, and the *N*-dependence of *D* in the long-chain regime increases, finally attaining $D∼N^{-2.2}$.

The authors remark that they do not expect the Rouse modes to be exact eigenmodes in their systems, but they still investigated $\langle X_{p}\left(t\right)X_{p}\left(0\right)\rangle$. The dependence of mean-square Rouse amplitudes $\langle {\left(X_{p}\left(0\right)\right)}^{2}\rangle$ on *p* and the two chain stiffness parameters was examined for an N=35 polymer. While increasing $k_{\theta }$ has little effect on $\langle {\left(X_{p}\left(0\right)\right)}^{2}\rangle$, for larger values of *p*, increasing $k_{\phi }$ considerably reduces $\langle {\left(X_{p}\left(0\right)\right)}^{2}\rangle$ by close to five-fold for p=20. Time correlation functions $\langle X_{p}\left(t\right)X_{p}\left(0\right)\rangle$ were found to follow stretched exponentials in time, with β and the mode relaxation time decreasing with increasing *p* and with an increase in chain stiffness.

Moreno and Colmenero [26] reported an extended simulation of an A-B blend of bead–spring polymers. The polymers were all shorter than the known nominal entanglement length, which is approximately 32 monomer beads. Beads were connected by a FENE potential and a monomer–monomer potential
(33)$$V_{\alpha \beta }\left(r\right)=4\epsilon ({\left(\frac{\sigma_{\alpha \beta }}{r}\right)}^{12}-7c^{-12}+6c^{-14}{\left(\frac{r}{\sigma_{\alpha \beta }}\right)}^{2},$$
with ϵ=1, c=1.15, and $\sigma_{\alpha \beta }$ = 1.6, 1.3, or 1, respectively, in units of $\sigma_{BB}$ for (α,β)=(A,A),(A,B), or (B,B), respectively, with a packing fraction 0.53 and a cutoff $\sigma_{\alpha \beta }$. Temperatures ranged from 1.5 to 0.33 in different simulations. All chains were the same length, with *N* between 4 and 21; systems contained 2000–4000 monomers. Dynamic asymmetry appeared because $\sigma_{\alpha \beta }$ differed between the two types of chain, with the A chains being larger and more numerous (70% of the total).

Moreno and Colmenero calculated the correlation functions $\langle X_{p}\left(0\right)X_{q}\left(t\right)\rangle$. For p≠q, $|\langle X_{p}\left(0\right)X_{q}\left(t\right)\rangle |\leq 0.1$ was found, which was taken to indicate that cross correlations between Rouse amplitudes are small. One notes that while the individual off-diagonal $\langle X_{p}\left(0\right)X_{q}\left(t\right)\rangle$ are small, for any *p* or *q*, there may be a respectably large number of them, so the total effect of the cross-correlations might not be negligible. The $\langle X_{p}\left(0\right)X_{p}\left(t\right)\rangle$ were found to be described by stretched exponentials in *t*. The stretching exponent β depends weakly on *p*, tending to decrease with increasing *p*. Considering chains with N=10, for the larger, more numerous A chains and T≥0.6, β∈(0.8,0.9). For the smaller, less numerous *B* chains, with decreasing temperature, β fell smoothly from 0.9 or so to 0.3 or so. That is, the relaxations are not the pure exponentials predicted by the Rouse model, but are closer to being single exponentials for the larger *A* chains, while being more remote from single exponentials for the *B* chains.

The two authors also considered how the measured $\tau_{p}$ scaled with N/p. For the A chains, except for the shortest-wavelength modes, $\tau_{p}$ scales approximately as ${\left(N/p\right)}^{2}$ at all temperatures studied. For N/p≤2, these being the shortest wavelength modes, $\tau_{p}$ tended to be slightly larger than a scaling line prediction. For the smaller, less numerous *B* chains, a normalized $\tau_{p}$ scaled as ${\left(N/p\right)}^{x}$, with x≈2.2 at high temperature but increasing to x≈3.5 at the smallest temperature studied. x≈3.5 matches the reptation prediction x=3, except, as the authors noted, they considered polymers that are too short to be entangled, indeed, polymers with as few as 4 monomer beads. For both chains, $N\langle {\left(X_{p}\left(0\right)\right)}^{2}\rangle$ scaled as ${\left(N/p\right)}^{2.2}$, which is not far from the Rouse value x=2. For chains of length N=15, the authors computed the mean-square displacement $\langle {\left(\Delta r\left(t\right)\right)}^{2}\rangle$ of the center beads of the B chains as a function of time at various temperatures. For times shorter than the Rouse time, there is at each temperature a region in which $\langle {\left(\Delta r\left(t\right)\right)}^{2}\rangle ∼t^{y}$, with a *y* that decreases markedly as the temperature is lowered. However, as the authors note, the entanglement crossover is reached by making polymer chains long, while here a crossover with the same appearance is reached by increasing the dynamic asymmetry, the value of $\sigma_{AA}/\sigma_{BB}$ between two components, neither of which is entangled. The authors note suggestions that these anomalous diffusive behaviors arise because density fluctuations around each chain become slow, but slowness “*may be induced by entanglement, but data reported here for the fast component suggest that this is not a necessary ingredient*”. They emphasize that, with sufficient dynamic asymmetry, entanglement-like dynamics are observed, even for model bead–spring tetramers.

Brodeck et al. [34] reported simulations of a polyethylene oxide melt and a comparison with inelastic neutron scattering studies. Simulations used Materials Studio 4.1 and Discover-3 (version 2005.1) with the COMPASS force field. In addition to potential energy terms reflecting bond stretching, bond bending, and bond torsion, their potential energy calculations include the coordinate cross-coupling terms known to be essential for calculating infrared and Raman frequencies. The polyethylene oxide oxygen has a significant partial charge, leading to Coulombic interactions. A Lennard–Jones 6–9 potential was used for the general non-bonded interaction. The simulation cell included five polymer chains, each composed of 43 monomer units in a 24.7Å cubic cell at temperatures 400, 375, and 350 K. Brodeck et al. made a series of tests of implications of the Rouse model, finding that the available scattering data on this system all agree with predictions of the Rouse model, namely that (i) for q ≤ 0.6 ${Å}^{-1}$, the characteristic relaxation time scales as $q^{-4}$, (ii) the incoherent scattering function depends on time as $\exp(-\alpha t^{1/2})$, at least for times longer than 2 pS, and (iii) the Rouse rate $W_{\mathrm{eff}}$ from simulations agrees with the experiment.

The Rouse mode amplitudes $X_{p}\left(t\right)$, their mean-square values, and their temporal autocorrelation functions were determined for 1≤p≤43. A test of orthogonality found that $\langle X_{p}\left(0\right)X_{q}\left(0\right)\rangle$ was ≪1, except for some large mode numbers. For each *p*, $\langle X_{p}\left(0\right)X_{p}\left(t\right)\rangle$ was described well by a stretched exponential in time. For small *p*, $\langle X_{p}\left(0\right)X_{p}\left(t\right)\rangle$ followed the Rouse model behavior so that $\langle {\left(X_{p}\left(0\right)\right)}^{2}\rangle$ and the integrated average relaxation time $\tau_{p}$ each scaled as $p^{-2}$. For larger p>8, $\tau_{p}$ decreased relative to an expected $p^{-2}$ dependence, finally reaching perhaps 2/3 of the expected value, while, over the same range of *p*, $\langle {\left(X_{p}\left(0\right)\right)}^{2}\rangle$ fell to a quarter of its expected value. $\langle X_{p}\left(0\right)X_{p}\left(t\right)\rangle$ was never a simple exponential. For p=1, the stretching exponent β was as large as 0.9. With increasing *p*, β falls, reaching a minimum of 0.7 or so for N/p close to 1, and then increasing slightly as N/p=1 is reached. β is weakly temperature dependent, especially at large *p*, increasing by a few percentage points as the temperature is increased.

Lahmar et al. [67] extended the earlier work of Tzoumanekas et al. [68] to consider the dynamics of their polymer model. In their model, individual beads represent 20-carbon backbone segments. Atomistic simulations were then used to determine the bead–bead nearest-neighbor-intrachain and interchain radial distribution functions and the three-bead angular distribution function. From these, the potentials of average force and thence, via an iterative process, the mean forces between polymer beads were determined. Bead motions were obtained using dissipative particle dynamics. Because the polymer beads are soft and can interpenetrate, a short range segmental repulsion force adequate to greatly reduce chain crossing was superposed. Chains with 6 to 40 beads, i.e., 120 to 800 carbon atoms in the backbone, were then examined. The center-of-mass diffusion coefficient was found to depend on chain length as $N^{-1.4}$ for the shortest chains and $N^{-2.3}$ for the longest chains, these values not being in agreement with the Rouse and reptation model predictions $N^{-1}$ and $N^{-2}$, respectively, though the latter is in reasonable agreement with experiments on polymer melts. The end-to-end vector reorientation time also depends on the chain length, its short- and long-chain dependencies being in approximate agreement with Rouse and reptation model predictions.

Lahmar et al. [67] examined the Rouse amplitude time correlation functions $\langle X_{p}\left(0\right)\cdot X_{p}\left(t\right)\rangle$, finding that these decay as stretched exponentials in time, not the pure exponentials required by the Rouse model. The stretching exponent β depends on *p* and on the chain length. For chains that cannot cross, β was smallest for *p* in the range N/2 to N/4. The authors propose that this length scale corresponds to the length scale of the network mesh as discussed by Tzoumanekas et al. [68]. For the longest chain studied, N=40, β decreased from 0.9 for the smallest and largest possible values of *p* to p≈2/3 at its minimum. When chain-crossing constraints were removed, β was close to unity at small *p* and decreased to 0.9 or so at large *p*. The mode relaxation times do not scale with *N* and *p* as predicted by the Rouse model. In the absence of the segmental repulsion chain-crossing barrier, the mode relaxation times do not depend on chain length, but are slower than expected at large *p*. In the presence of chain-crossing constraints, for large *N*, the relaxation times only follow reptation model predictions for the first few values of *p*. Lahmar et al. concluded that Rouse modes are not the system’s normal modes but did not claim that the system actually has normal modes of relaxation.

Perez-Aparicio et al. [69] reported a molecular dynamics simulation of poly(ethylene-*alt*-propylene) based on a coarse-grained bead and spring model. Their coarse-grained potentials were the bead–bead potentials of average force determined by comparison with simulations made on shorter chains using an atomistic potential, with a single coarse-grained bead representing ten monomers, about half of the nominal entanglement length. Coarse-grained chains containing between 5 and 30 beads, with 150 to 80 chains in a simulation cell, were then simulated. A coarse-grained friction factor *f* was set so that the long-time mean-square center-of-mass displacement of the coarse-grained short chains matched the long-time mean-square center-of-mass displacement of the atomistic chains. The short-time mean-square center-of-mass and bead displacements of the coarse-grained and atomistic chains do not match, apparently because for the coarse-grained chains, a short-time frictional memory function is replaced with a simple friction factor.

The authors then studied chain dynamics by calculating the $\langle X_{p}\left(0\right)X_{p}\left(t\right)\rangle$, which were found to follow stretched exponentials in time. The dependence of the mean-square amplitude $\langle {\left(X_{p}\left(0\right)\right)}^{2}\rangle$, the stretching parameter $\beta_{p}$, an average relaxation time $\tau_{p}$, and a Rouse frequency $W_{\mathrm{eff}}$ on mode number *p* and chain length *N* were considered. $W_{\mathrm{eff}}$ is the inverse of a nominal time needed for a bead to diffuse through the length of a statistical segment. For *p* close to unity, relaxations were very nearly exponential, with $\beta_{p}\approx 1$. With increasing *p*, $\beta_{p}$ fell, declining to $\beta_{p}\approx 0.8$ for N/p in the range 2–3. At still larger *p*, $\beta_{p}$ increased again, reaching 0.95 or so for N/p≈1. For N/p>5, $\langle {\left(X_{p}\left(0\right)\right)}^{2}\rangle$ and $\tau_{p}$ followed the Rouse model predictions; at smaller N/p, $\langle {\left(X_{p}\left(0\right)\right)}^{2}\rangle$ and $\tau_{p}$ are both substantially smaller than the Rouse predictions. The authors indicate their simulations are unrealistic at very small N/p, roughly 1<N/p<1.4, this regime being much narrower than the N/p<5 regime in which the Rouse behavior is not seen. The Rouse frequency $W_{\mathrm{eff}}$ depends strongly on N/p, increasing perhaps fourfold as *p* is increased from 1 to its upper limiting value N−1. When chains are allowed to pass through each other, the dependence of $W_{\mathrm{eff}}$ on *p* is reduced by approximately 50%, while the modest dependence of $W_{\mathrm{eff}}$ on *N* disappears. These features are not consistent with the hypothesis that the Rouse model provides a valid description of polymer dynamics in the melt.

Perez-Aparicio et al. [69] repeat the warnings of Akkermans, Padding, and Briels [20,54,70,71] that the coarse-graining of atomic coordinates leads to friction forces, ‘random’ thermal forces, and can permit long polymer chains to pass through each other if appropriate precautions are not taken. The rigorous statistico-mechanical representation of coarse graining is provided by the Mori–Zwanzig formalism [72], in which the complete set of atomic coordinates is partitioned between variables retained for study and variables described as *bath variables*, the latter being subject to a thermal averaging process that removes them from further consideration. The Mori–Zwanzig coarse-graining process introduces to the equations of motion a set of random *thermal* forces, a corresponding set of frictional forces that together with the thermal forces keep the system in thermal equilibrium, and a set of Mori memory kernels ϕ(t) that replace friction factors, namely
$$-fv\left(t\right)\to -\int_{-\infty }^{t}ds\;\phi \left(t-s\right)v\left(s\right).$$

Here, v(t) is a bead velocity at time *t*. One could, in principle, use simulations to recover the Mori memory kernels ϕ(t−s), as was done in a different class of systems by Phillies and Stott [73,74].

Perez-Aparicio et al. [69] explore possible paths for uniting their simulations with the Rouse model. They note the possibility that the statistical segment length *b* or the friction constant *f* could depend on the mode so that a mode dependence of *b* or *f* might explain some of their results. In the context of the Rouse model, *b* and *f* are associated with individual bonds or beads, respectively, so it is unclear how a simple *p*-dependent form of these variables could be interpreted, other than as a formal re-parameterization. However, each $\langle X_{p}\left(0\right)X_{p}\left(t\right)\rangle$ samples *t* with slightly different weightings for each time, and therefore represents a different averaging over a memory function ϕ(t−s), thus providing a mechanism that would lead to formal *p* dependencies of *b* and *f*. Some authors noted that united atom simulations show different short-time dynamics than do all-atom simulations. Failure to treat friction with a memory function ϕ(t−s), as required by the Mori [29,30] formalism, rather than with a simple friction factor *f*, would lead to this outcome. The extensive review by Jin et al. [75] discusses current results on applying approximations to the Mori–Zwanzig formalism to molecular dynamics in the presence of united atoms or coarse graining.

Kalathi et al. [17] reported simulations of melts of 500–2000 linear polymer chains, with individual chains containing between 10 and 500 beads. The objective was to analyze polymer motions in terms of Rouse modes, the hope being that the dynamics of longer- and shorter-wavelength (smaller and larger *p*) Rouse modes would reveal the motions of longer and shorter chain segments. As a justification for the study, experimental and simulational evidence was cited for deviations from Rouse behavior at short distances and small times [34,76]. Temporal autocorrelation functions of Rouse amplitudes were found to decay as stretched exponentials in time, not the simple exponentials predicted by the Rouse model [77], with mean-square amplitudes that fail to scale as $p^{-2}$. Additionally, the stretching parameter β was previously found by Padding and Briels to depend on *p* [20,54]. Finally, Kalathi et al. noted the result of Likhtman [16] that Rouse modes are sometimes substantially cross correlated (${\langle X_{p}\left(t\right)X)}_{q}\left(0\right)\rangle \neq 0$ for p≠q and t>0).

In Kalathi et al.’s simulations, the bead–spring Kremer–Grest model was used, with a finitely extensible nonlinear-elastic potential for bonded beads, a Lennard–Jones interaction between all pairs of unbonded beads, and a bending potential between linked segments along each chain. This potential energy keeps chains from passing through each other. In some simulations, the potential energy was modified to permit chain crossing, the modified potential energy being such as to not change chain static properties significantly, thus testing for effects arising from topological interactions. Rouse amplitude time correlation functions $\langle X_{p}\left(0\right)X_{p}\left(t\right)\rangle$ were evaluated for *p* as large as 24 and 4 values for the chain stiffness. Simulations were executed for sufficiently large *t* such that $\langle X_{p}\left(0\right)X_{p}\left(t\right)\rangle$ decayed to essentially zero. Except at the earliest times, where in some systems for small *p* the fitted curves pass above the data, their $\langle X_{p}\left(0\right)X_{p}\left(t\right)\rangle$ measurements were described well by stretched exponentials in time.

From the simulations, $\langle X_{p}^{2}\left(0\right)\rangle \sin^{2}\left(p\pi /2N\right)$ fails to be independent of N/p, contrary to the prediction of the Rouse model. This product does approach the characteristic ratio asymptotically for small *p*. With decreasing N/p, the product $\langle X_{p}^{2}\left(0\right)\rangle \sin^{2}\left(p\pi /2N\right)$ decreases slowly until N/p reaches 15 or so; at smaller N/p, this function decreases rapidly toward 0.5–0.8 at N/p=2. The stretching parameter β of $\langle X_{p}\left(0\right)X_{p}\left(t\right)\rangle$ also depends on N/p, being roughly 0.9 at p=1, decreasing to a minimum β≈0.5 for N/p≈40, and then increasing to 0.75 or 0.8 for N/p≤5. The minimum in β is found for $N/p\approx N_{e}$, $N_{e}$ being an inferred entanglement length. β also depends modestly on the bending constant in the force field.

Kalathi et al. reported results for a range of chain lengths, from N=10 to N=500. At fixed N/p, $W^{\mathrm{eff}}$ is independent of *N* for shorter chains, but for chains with N≥150 and N/p>15, the authors report that $W^{\mathrm{eff}}$ decreases markedly with increasing N/p. β also shows distinct behaviors for shorter and longer chains. For N≤100, β simply increases as N/p is increased. At larger *N*, β shows a minimum at intermediate N/p. Kalathi et al. reported the effective monomeric relaxation rate $W^{\mathrm{eff}}$ and the longest relaxation time. $W^{\mathrm{eff}}$ increases by ten- to thirty-fold between small and large *p*.

Removing the chain-crossing constraint considerably increases $W^{\mathrm{eff}}$, especially for longer chains, eliminates the dependence of $W^{\mathrm{eff}}$ on chain length, and eliminates the dip in β at intermediate N/p. In the absence of the chain-crossing constraint, β increases monotonically from 0.8 at smaller N/p to near unity as p→1, while the longest relaxation rate increased as $N^{2}$ for all chain lengths studied, rather than—in the presence of chain uncrossability—increasing more rapidly than $N^{2}$ at large *N*.

The central conclusion of Kalathi et al.’s simulations is that the Rouse amplitudes of their bead–spring polymers show one behavior if the chains are short and a different behavior if the chains are longer than some length $N_{e}$, but the short chain–long chain difference disappears if the chains are enabled to pass through each other. However, removing the chain-crossing constraint does not replace the stretched-exponential time dependencies of $\langle X_{p}\left(0\right)X_{p}\left(t\right)\rangle$ with the simple-exponential time dependence of the Rouse model. Their results provide direct evidence for the contribution of topological interactions (chain-crossing constraints) to the dynamics of long polymers.

Hsu and Kremer [78,79] made molecular dynamics simulations of 1000 chains containing N = 500, 1000, or 2000 beads at volume fraction 0.85, using a novel scheme for equilibrating a large assembly of long chains [80]. The chains were described by a bead–spring model [37] with a FENE potential between bonded atoms, a Lennard–Jones potential between non-bonded atoms, and in different simulations, a bend-bonding constant $k_{\theta }$ of 0 or 1.5. Molecular dynamics were executed using the ESPResSo++ package. Averaging, especially of the stress tensor $\sigma_{\alpha \beta }$, was improved by making an ensemble as well as a time average, namely by averaging over ten independently generated equilibrated systems as well as an extended time average. The nominal distance $N_{e}$ between the hypothesized entanglement points was estimated from primitive path analysis, and from Green–Kubo determinations of viscoelastic properties, to be 26–28 monomers, so the 1000-bead chain would nominally have contained three dozen entanglement lengths and proportionately more or less for the 2000 and 500 bead chains.

Hsu and Kremer [78] first calculated static correlation functions, including the distribution functions for the end-to-end distance and the radius of gyration, the mean-square distance between pairs of polymer beads as a function of the distance between them along the chain, the bond–bond orientation correlation function, and the static structure factor S(q), thus identifying a good compromise value for the bond-bending constant. They then developed several dynamic properties, beginning with the time dependence of mean-square displacements of inner monomers, of inner monomers with respect to the center of mass, and of the center of mass itself. Comparison was made with power-law time dependencies. They made a primitive path analysis based on cooling the chains while holding the ends fixed, and computed the stress relaxation modulus from the stress tensor.

Hsu and Kremer [79] advanced to study the $\langle X_{p}\left(t\right)X_{p}\left(0\right)\rangle$ and $g^{\left(1\right)}\left(q,t\right)$. For each *p*, $\langle X_{p}\left(0\right)X_{p}\left(t\right)\rangle$ was found to decay as a stretched exponential in *t*, relaxations being longer lived at larger N/p. The deviation from a simple-exponential decay was smaller for large N/p, i.e., for small *p*, and was less obvious at early times. For small N/p, say N/p≤25, i.e., p≥N/25, $\langle X_{p}\left(0\right)X_{p}\left(t\right)\rangle$ was found to be independent of the chain length. For larger N/p, $\Phi_{ppii}\left(t\right)$ was seen to depend on *N*, $\Phi_{ppii}\left(t\right)$ decaying less rapidly for the longer chains. The parameters of the stretched exponential were found to depend on N/p. The stretching exponent β decreases from 0.8 or so at small N/p to a minimum near 0.5 for N/p near 25 or 80 (depending on the chain stiffness $k_{\theta }$), and then increases to 0.6 or so for large N/p. The location of the minimum in β is independent of the polymer length. The value of N/p for which β reaches its minimum is approximately the same as the value of N/p above which $\Phi_{ppii}\left(t\right)$ begins to depend on *N*. The average time constant τ increases through eight orders of magnitude as N/p is increased from 1 to 1000. Hsu and Kremer compare τ to power laws ${\left(N/p\right)}^{x}$, using asserted theoretical model values x=2 for small N/p and x=3.4 for large N/p. Regions where these predictions are approximately correct are indeed found, though, at large N/p, τ tends to roll off to below the x=3.4 power law curve. Hsu and Kremer propose that the stretched-exponential behavior arises as a crossover from Rouse behavior at early times to reptational behavior at longer times.

Finally, Hsu and Kremer [79] discussed the coherent and incoherent structure factors $g^{\left(1\right)}\left(q,t\right)$ and $g^{\left(1s\right)}\left(q,t\right)$. $g^{\left(1s\right)}\left(q,t\right)$ is the self part of the sum that defines $g^{\left(1\right)}\left(q,t\right)$. Their analysis is based on the self part of the Gaussian approximation
(34)$$g^{\left(1s\right)}\left(q,t\right)=\langle \sum_{i=1}^{N}\exp\left(ıq\cdot \left(r_{i}\left(t\right)-r_{i}\left(0\right)\right)\right)\rangle \approx \sum_{i=1}^{N}\exp\left(-\frac{q^{2}}{6}\langle {\left(r_{i}\left(t\right)-r_{i}\left(0\right)\right)}^{2}\rangle \right)$$
where $\langle {\left(r_{i}\left(t\right)-r_{i}\left(0\right)\right)}^{2}\rangle$ is the mean-square displacement of bead *i* during time *t*. Hsu and Kremer report $\log(g^{\left(1\right)}\left(q,t\right))$ and $\log\left(g^{\left(1s\right)}\left(q,t\right)\right)/q^{2}$ as functions of time for a half-dozen values of *q*. If the Gaussian approximation is correct, the latter quantity would be independent of *q*. Contrariwise, if $\log\left(g^{\left(1s\right)}\left(q,t\right)\right)/q^{2}$ has a pronounced *q*-dependence, this quantity would be determined by the all moments ${\left(r_{i}\left(t\right)-r_{i}\left(0\right)\right)}^{2n}$, n≥1 of the bead displacement, not just the mean-square bead displacement [81]. Indeed, Hsu and Kremer reported that at short times, $\log\left(g^{\left(1s\right)}\left(q,t\right)\right)/q^{2}$ is independent of *q*, so at short times, the Gaussian approximation is correct and $\log\left(g^{\left(1s\right)}\left(q,t\right)\right)/q^{2}$ gives the mean-square bead displacement.

At times longer than a hypothesized entanglement time $\tau_{e}$, $\log\left(g^{\left(1s\right)}\right)/q^{2}$ depends markedly on *q*, so in the hypothesized reptation regime, the Gaussian approximation fails, in which case $\log\left(g^{\left(1s\right)}\left(q,t\right)\right)/q^{2}$ does not reveal the mean-square displacement of the beads and does not provide a test of the existence of the proposed $t^{1/4}$ motional regime. The hypothesized power laws from the reptation model were drawn as tangent lines to some of the $\log\left(g^{\left(1s\right)}\left(q,t\right)\right)/q^{2}$ curves, transitions between the hypothesized motional regimes being estimated as points where the power-law lines intersect. From the estimated times, an estimate of the entanglement length $N_{e}$ was obtained, this length being approximately in agreement with $N_{e}$ estimated from primitive path analysis or from melt viscoelastic properties.

Goto et al. [82] used the Kremer–Grest bead–spring model to study melts of linear and ring molecules. Their calculations incorporated 10,000 beads, linked in different simulations into molecules having between 5 and 400 linked beads, beads interacting via FENE, bending angle, and Lennard–Jones potentials. The temporal autocorrelation functions of the Rouse amplitudes were calculated. For each molecular weight and *p*, $\langle X_{p}\left(0\right)X_{p}\left(t\right)\rangle$ was found to be described well by a stretched exponential in time. The stretching exponent β, the mean-square amplitude $\langle {\left(X_{p}\left(0\right)\right)}^{2}\rangle$, and the relaxation time τ were all found to depend on the mode number *p* and the length *N* of the polymer chain as functions of the unified variable N/p. For ring polymers, β was never larger than 0.8, with a local minimum of 0.7 near N/p≈4, a return to 0.8 for N/p≈20, and a tendency to decrease at larger N/p, contrary to the Rouse model prediction β=1 at all N/p. For linear polymers, β was 0.85 at N/p=1, decreased to 0.7 for N/p≈2, increased to 0.8 or so near N/p≈5, and for N/p≈100 decreased again to 0.6 or so. According to the Rouse model, the normalized Rouse amplitude $\sin^{2}\left(\pi p/N\right)\langle {\left(X_{p}\left(0\right)\right)}^{2}\rangle$ should be a constant independent of N/p. For both rings and linear chains, in Goto et al.’s simulations, this quantity instead increases roughly five-fold between N/p≈1 and N/p≈30; for ring polymers at N/p>30, this quantity clearly declines again with increasing N/p. The integrated relaxation time τ increases with increasing N/p, approximately as a power law in N/p, the power being close to the Rouse model value 2 at smaller N/p and larger, especially for the linear polymers, at large N/p. Goto et al. described this N/p dependence as “*the deviation from the model behavior becomes remarkable…*”. The exponent of the power law visibly increases smoothly with increasing N/p, without a sharp transition in its value at some N/p.

Goto et al. [82] also examined P(r,t), the probability distributions for the displacement of single beads and for chain centers of mass. For ring polymers, P(r,t) remains nearly Gaussian at all times. For linear polymers, at larger times, P(r,t) deviates from the expected Gaussian form, becoming broader as *t* increases. Characterizing the deviation from a simple Gaussian with a non-Gaussian parameter $\alpha_{2}\left(t\right)$,
(35)$$\alpha_{2}\left(t\right)=\frac{3\langle {\left(R\left(t\right)\right)}^{4}\rangle -5{\langle {\left(R\left(t\right)\right)}^{2}\rangle }^{2}}{5{\langle {\left(R\left(t\right)\right)}^{2}\rangle }^{2}},$$
$\alpha_{2}\left(t\right)$ for monomers in linear chains has a small short-time peak, roughly the same size for all chain lengths, and a long-time peak whose size increases perhaps linearly with the chain length. For monomers in ring polymers, only the small short-term peak is apparent.

Roh et al. [12] simulated ring and linear polyethylenes that had short-chain branches spaced along their lengths. The polyethylene backbones each contained 400 carbon atoms, with 33 side chains, each being 5 carbon atoms in length; as a control, simulations were made of polymers having the same backbones but no side chains. Simulated systems contained 54–70 molecules held at constant temperature and pressure 450 K and 1 atm. Molecules interacted via TraPPE united atom potentials, motions being calculated with the r-RESPA algorithm [83,84,85]. The addition of short chains to the polymers renders the molecules more compact.

Rouse mode amplitudes $X_{p}\left(t\right)$ were computed. For linear chains, the equilibrium average $\langle {\left(X_{p}\left(0\right)\right)}^{2}\rangle$ was linear in $N/p^{2}$ for large $N/p^{2}$, but, for $N/p^{2}\leq 10$, $\langle {\left(X_{p}\left(0\right)\right)}^{2}\rangle$ fell below the linear line. For ring polymers, $\langle {\left(X_{p}\left(t\right)\right)}^{2}\rangle$ had a linear dependence on $N/p^{2}$, with a slight drop-off from that dependence in the limits of small or large $N/p^{2}$. Roh et al. also examined the temporal autocorrelation functions $\langle X_{p}\left(0\right)X_{p}\left(t\right)\rangle$. In all cases examined, $\langle X_{p}\left(0\right)X_{p}\left(t\right)\rangle$ deviated from a simple exponential decay, having instead the form of a stretched exponential in time. Writing the stretched exponential as $\exp(-{\left(t/\tau_{p}\right)}^{\beta })$, $\tau_{p}$ was found to scale as ${\left(N/p\right)}^{2}$.

A few studies used very different bead–bead potential energies. Smith et al. [86,87] used a tangent hard-sphere polymer model, in which the bond length between adjoining spheres was allowed to vary over a narrow range. Their polymer chains had any of the 6 lengths between 8 and 192 spheres, at volume fractions 0.3, 0.4, or 0.45. Systems contained 32 or 64 chains in a cell with periodic boundary conditions. The simulations covered five orders of magnitude in time. Smith et al. studied the behavior of static and dynamic properties, including chain dimensions, Rouse amplitudes, chain mean-square displacements, fluctuations in the chain end-to-end vector’s length and orientation, evidence for knot formation, dynamic structure factors, and Rouse amplitude autocorrelation functions. Smith et al. asserted that reptation and the tube model predict that, at times shorter than the entanglement time, for chain segments shorter than the mean number of beads between two entanglements, the Rouse model should be applicable, but at long times, the Rouse mode relaxation rates should increase in proportion to the number of times that a typical chain is entangled.

From the amplitudes $X_{p}\left(t\right)$ of the Rouse mode, Smith et al. [86,87] calculated the normalized temporal correlation functions $\langle X_{p}\left(0\right)X_{p}\left(t\right)\rangle /\langle {\left(X_{p}\left(0\right)\right)}^{2}\rangle$. These were plotted against the reduced times $tp^{2}/N^{2}$, where *p* is the mode number and *N* is the chain length. For a 32-bead polymer at volume fraction 0.3, the correlation functions are very nearly exponential. At the larger polymer concentrations, the higher-order modes (2≤p≤6) deviate from simple-exponential behavior, their relaxations being slower than exponential at long times, with relaxation rates scaling linearly in $p^{2}$. For a 192-bead polymer, relaxations are markedly non-exponential, the deviation from single exponential behavior being most notable at the largest polymer volume fraction. In contrast to the 32-bead polymer, for the 192-bead polymer, the non-exponential behavior is most prominent for small *p*. Additionally, for the 192-bead polymers, the mode relaxation rates are not linear in $p^{2}$. At ϕ=0.45, the correlation function for p=1 gains a pronounced shoulder, after which it decays very rapidly toward zero. The Rouse model prediction of exponential decay of $\langle X_{p}\left(0\right)X_{p}\left(t\right)\rangle$ is thus not sustained for longer polymer chains.

Smith et al. reported seeing Rouse behavior, based on the observations $D∼N^{-1}$ for shorter polymers and mean-square displacement proportional to $t^{1/2}$ at shorter times. Their *D* against *N* plot shows, for each polymer volume fraction, a smooth curve approaching a tangent to $D∼N^{-1}$ at small *N* and to $D∼N^{-2}$ at large *N*, though most of their points are in the intermediate transition region. At intermediate times, they found mean-square displacements $∼t^{0.28}$ or $t^{0.31}$. $t^{1/4}$ behavior was not observed. They also noted an anomaly: Following a second $t^{1/2}$ regime in the mean-square displacements, they observed a small plateau in the time dependencies of the mean-square displacement and the end-to-end vector time correlation function. They proposed that the plateau corresponds to a time-scale separation between two types of entanglements, namely tight local knots and surrounding chains functioning as fixed obstacles.

Can Rouse coordinates and modes emerge naturally from an analysis of Brownian or molecular dynamics simulations? This question was approached by Wong and Choi [88], who considered a non-linear bead–spring model, in which individual terms of the linked-bead potential have a form
(36)$$U_{ij}=\frac{1}{2}k(|r_{i}-r_{j}{\left|-b\right)}^{2}.$$

Here, *k* is a force constant, $r_{i}$ is the position of bead *i*, and *b* is a constant. This potential energy term has a minimum when beads *i* and *j* are separated by the distance *b*. In the original Rouse model, b=0; when in their rest positions, all beads of a Rouse chain are at the same location. For their Brownian dynamics simulations, Wong and Choi integrated their equations of motion with the Newton–Gauss method. Wong and Choi also made molecular dynamics simulations of chains in a melt, using the TraPPE [84,85] force field with GROMACS [41] as an integrator. For the molecular dynamics simulations, individual chains had 30–73 beads, the number of chains being fixed at 40 for simulations of longer chains.

For each molecule and time step, Wong and Choi then calculated, separately along each Cartesian coordinate, the distance $q_{i}^{\prime }$ for each bead *i* from the chain center of mass. They then formed and averaged the N×N single-time correlation matrices $C_{i,j}$
(37)$$C_{i,j}=\frac{1}{S}\sum_{t=0}^{S}q_{i}^{\prime }\left(t\right)q_{j}^{\prime }\left(t\right),$$
*t* here being the label on the *S* time steps. Proper orthogonal decomposition was then used to obtain dominant eigenvectors of $C_{i,j}$. Simulations of single chains using Brownian dynamics and simulations of chain melts using molecular dynamics yielded similar forms for the eigenvectors of $C_{i,j}$. Wong and Choi examined linear, ring, and star polymers. The dominant eigenvectors of the linear polymer closely resemble the eigenvectors of the Rouse model.

The temporal autocorrelation functions of the eigenvectors were then evaluated, finding that the autocorrelation functions decay as stretched exponentials in time, with values of β close to unity. For ring and star polymers, two or three, respectively, of the relaxations were found to be nearly degenerate in their relaxation times. The temporal cross-correlation functions were not evaluated. The integrated average relaxation time for the modes scales differently for the Brownian dynamics and the molecular dynamics simulations, namely $\tau ∼{\left(N/p\right)}^{2}$ for the Brownian dynamics simulations but $\tau ∼{\left(N/p\right)}^{2.3}$ for the molecular dynamics simulations.

## 5. Discussion

In this section, we gather together results from different studies that tend to present a coherent description of simulational tests of the Rouse model. The model has two assumptions that precede any demonstration of the Rouse modes and their properties.

First, there is a distribution function for the likelihood of finding a pair of polymer beads separated in space by a distance *r*. In the model, that distribution function is a Gaussian in *r*. Correspondingly, the potential of average force between the two beads is quadratic in *r*, leading to a Hooke’s law restoring force that pulls the beads toward each other.

Second, the random thermal forces on the polymer beads, due to their interactions with neighboring polymer molecules, are described by independent Gaussian random processes. The correlation times of the thermal forces are taken to be sufficiently short such that the thermal forces on a bead at a series of times can be treated as independent of each other. As a result, the displacements of beads have Gaussian distributions, as seen in the simplest case for the Brownian motion of an isolated bead monomer. For the intermediate structure factor $g^{\left(1s\right)}\left(q,t\right)$, the Gaussian approximation
(38)$$g^{\left(1s\right)}\left(q,t\right)\equiv \langle \exp(-ıq\cdot \left(r_{i}\left(t\right)-r_{i}\left(0\right)\right))\rangle =\exp\left(-q^{2}\langle {\left(r_{i}\left(t\right)-r_{i}\left(0\right)\right)}^{2}\rangle \right)$$
is, therefore, exact, not just an approximation.

These assumptions are not uniformly sustained by the simulations.

First, several authors simulated the intramolecular atom–atom radial distribution functions. Tsalikis et al. [13] found that the bead–bead static distribution functions were in general not Gaussians, even though the calculated static structure factors were in good agreement with the experiment. On the other hand, Papadopolous et al. [40] reported that the bead–bead distances are Gaussianly distributed.

Second, if the Gaussian approximation for the center-of-mass motion is exact, which is implicit in the Rouse model, then the mean-square center-of-mass displacement must increase linearly with time. Kopf et al. [14], Krushev et al. [51], and Paul et al. [43] instead reported that they found subdiffusion (mean-square center-of-mass displacement proportional to $t^{a}$ for a<1) on shorter time scales. Smith et al. [47] found that $g^{\left(1\right)}\left(q,t\right)$ calculated directly in their simulations did not agree with $g^{\left(1\right)}\left(q,t\right)$ calculated from the measured mean-square bead displacements and the Gaussian approximation, and concluded that the Gaussian approximation was invalid in their system. By implication, reports of mean-square center-of-mass displacements, when inferred from $g^{\left(1\right)}\left(q,t\right)$ by applying the Gaussian approximation, require additional validation.

Third, with computer simulations, one can determine the distribution function P(r,t) for bead displacements *r* through times *t*. Goto et al. [82] did this, showing directly that P(r,t) for linear but not ring polymers becomes non-Gaussian at longer times.

Rouse modes are predicted by the Rouse model to have a series of properties, with the following in particular.

First, $\langle X_{p}\left(0\right)X_{p}\left(t\right)\rangle$ is predicted to decay in time as a simple exponential. Contrary to this prediction, $\langle X_{p}\left(0\right)X_{p}\left(t\right)\rangle$ shows a stretched exponential dependence on time as seen in the results of Abrams et al. [55], Bulacu and van der Giessen [64], Goto et al. [82], Hsu et al. [79], Kalathi et al. [17], Krushev et al. [51], Kremer and Grest [37], Lahmar et al. [67], Papadopolous et al. [40], Perez-Aparicio et al. [69], Roh et al. [12], Tsalikis et al. [13], and Tsolou et al. [11]. In addition, several studies, including simulations by Brodeck et al. [34], Harmandaris et al. [48], and Tsolou et al. [58], reported that $\langle X_{p}\left(0\right)X_{p}\left(t\right)\rangle$ is not a simple exponential in time but did not identify its actual functional form. As a function of *p*, β is found to have a minimum at some intermediate value of *p* but to be larger for *p* near 1 or near *N*.

We emphasize that there are two sets of related results that do support the validity of the Rouse model. First, Paul et al. [43] found that $\langle X_{p}\left(0\right)X_{p}\left(t\right)\rangle$ indeed does relax exponentially, at least for p≤3. Second, Smith et al. [86,87] concluded that they observed Rouse-like behavior, based on finding $D∼N^{-1}$ and mean-square displacements $∼t^{1/2}$. However, cf. the previous paragraph, a considerable number of more recent studies found stretched-exponential, not simple-exponential, relaxations.

Several authors have explored the possible causes of the observed non-exponential relaxations. Padding and Briels [54] introduced a switching scheme that permits the chain crossing constraint to be turned on or off. Turning on the constraint and adding a bond-bending potential causes β to fall from near unity to 0.77 for higher modes, and slows the relaxation of the p=1 or 2 modes. Lahmar et al. [67] found in the presence of chain-crossing constraints that β depended markedly on *p*, decreasing from β=0.9 for *p* close to one to a minimum of β≈2/3. When the constraint was removed, β was ≈1 at small *p* and never less than 0.9 as *p* was increased.

Second, the mean-square average $\langle {\left(X_{p}\left(0\right)\right)}^{2}\rangle$ is predicted to depend on *p* in such a way that $\langle {\left(X_{p}\left(0\right)\right)}^{2}\rangle \sin^{2}\left(p\pi /2N\right)$ is independent of *p*, so for small *p*, one has $\langle {\left(X_{p}\left(0\right)\right)}^{2}\rangle ∼p^{-2}$. Goto et al. [82] instead found that $\langle {\left(X_{p}\left(0\right)\right)}^{2}\rangle \sin^{2}\left(p\pi /2N\right)$ depends on *p*, changing by a factor of three over the observed range of *p*. Kalathi et al. [17] reported that this function decreases with decreasing N/p, declining to 0.5–0.8 by N/p≈2. Padding and Briels [20] confirmed the presence of this *p*-dependence and interpreted it as arising from chain stiffness. Bulacu and van der Giessen [64] considered the effect of bending and bending–torsion force constants on $\langle {\left(X_{p}\left(0\right)\right)}^{2}\rangle$; the former has little effect, while increasing the latter substantially reduces $\langle {\left(X_{p}\left(0\right)\right)}^{2}\rangle$.

Several authors reported the *p* dependence of $\langle {\left(X_{p}\left(0\right)\right)}^{2}\rangle$ without making a full comparison with the prediction that $\langle {\left(X_{p}\left(0\right)\right)}^{2}\rangle ∼\sin^{-2}\left(p\pi /2N\right)$. For small *p*, a $p^{-2}$ dependence is found, in agreement with the Rouse model [40]. At larger *p*, $\langle {\left(X_{p}\left(0\right)\right)}^{2}\rangle$ was found by Harmandaris et al. [48], Paul et al. [43], and Krushev et al. [51] to decrease as $p^{-3}$. The latter authors identified the $p^{-3}$ dependence with the behavior of a freely rotating chain. Tsolou et al. [11] found the $p^{-2}$ behavior for $N/p^{2}\geq 2$ and a stronger dependence for larger *p*, namely $N/p^{2}<2$. Brodeck et al. [34] reported $p^{-2}$ behavior for p≤8 but not for larger *p*. Doxastakis et al. [56] reported that $p^{-2}$ behavior is limited to p≤3. Moreno and Colmenero [26] simulated an A-B blend, finding different behaviors for the A and B chains.

In the Rouse model, modes are said to be orthogonal so that $\langle X_{p}\left(0\right)X_{q}\left(t\right)\rangle =0$ for p≠q. Kopf et al. [14], Paul et al. [43], Smith et al. [47], Padding and Briels [20], and Tsalikis et al. [13] confirmed this expectation for t=0. However, Likhtman [16] showed $\langle X_{1}\left(t\right)X_{3}\left(0\right)\rangle \neq 0$ but only for t>0.

No contradiction is seen here. The average that generates $\langle X_{p}\left(0\right)X_{q}\left(t\right)\rangle$ is not the same as the scalar product for the basis vectors of $X_{p}$ and $X_{q}$, so rationales based on the Schwarz inequality do not prove that $\langle X_{p}\left(t\right)X_{q}\left(0\right)\rangle$ must be zero at all times if $\langle X_{p}\left(0\right)X_{q}\left(0\right)\rangle =0$. Correspondingly, simulational tests for the orthogonality of Rouse modes must examine full time correlation functions, not merely their static (t=0) values.

Furthermore, there is an interesting analogy with spatial Fourier components $a_{k}$ of the density of a simple liquid. $\langle a_{k}\left(0\right)a_{-q}\left(0\right)\rangle$ must vanish by translational invariance if k≠q, but $\langle a_{k}\left(0\right)a_{q}\left(0\right)a_{-k-q}\left(0\right)\rangle$ corresponds to the three-body radial distribution function $g^{\left(3\right)}\left(r_{1},r_{2},r_{3}\right)$, which contains non-trivial information about the liquid. By analogy, even if $\langle X_{p}\left(0\right)X_{q}\left(t\right)\rangle =0$ for p≠q, the triple correlation function $\langle X_{p}\left(0\right)X_{q}\left(t\right)X_{p+q}\left(t^{\prime }\right)\rangle$ is not obliged to be uninteresting.

Finally, Tsalikis et al. [13] made the valuable and correct point that $\langle X_{p}\left(t\right)X_{p}\left(0\right)\rangle$ as displayed on a semilog plot ‘seems to be exponential-like’ (i.e., is close to a straight line), except at short times. This point does not contradict the observation that $\langle X_{p}\left(t\right)X_{p}\left(0\right)\rangle$ follows well a stretched exponential in time. On semilog plots, ‘exponential-like’ behavior is a general though not universal feature of stretched-exponential time dependencies at long times, namely if we have a function,
(39)$$y=\exp(-\alpha t^{\beta })$$
then its logarithmic derivative, the apparent slope on a semilog plot, is
(40)$$\frac{d\ln(y)}{dt}=-\alpha \beta t^{\beta -1}$$

For β≈0.8, as observed, the derivative becomes
(41)$$\frac{d\ln(y)}{dt}\approx -\frac{\alpha \beta }{t^{0.2}}.$$

At small *t*, this function diverges, implying that if one advances to small *t*, one has moved outside the stretched exponential’s domain of validity. At larger times, $t^{0.2}$ is nearly a constant, leading on a semilogarithmic graph to a function whose slope is nearly a constant, i.e., the function appears to be close to linear. However, the slope does depend on time. The apparent slope obtained from a linear fit to a section of a stretched exponential is then an artifact determined by the time interval, over which the fit is evaluated.

## 6. Conclusions

Summarizing from many papers, with respect to polymer dynamics and the Rouse model, we have the following.

First, the Rouse coordinates are just that, coordinates, generated from the Cartesian coordinates of the model polymer beads by means of a discrete Fourier transform. This is a purely mathematical replacement of one representation of the bead positions with another and has no physical content. Correspondingly, a challenge to the validity of the Rouse coordinates *as coordinates* is a challenge to the validity of Fourier’s theorem, so the challenge is highly unlikely to be correct.

However, the Rouse *coordinates* are sometimes also interpreted as the Rouse *modes*, which, within the Rouse model of a single isolated polymer chain, are predicted to have a series of physical properties. In a quiescent polymer melt, the observed Rouse modes do not have the predicted physical properties. In particular, we have the following:

(1) The relaxation $\langle X_{p}\left(0\right)X_{p}\left(t\right)\rangle$ of the temporal autocorrelation function of a single Rouse amplitude is a stretched exponential in time, not the pure exponential predicted by the Rouse model. At larger *p* (shorter wavelength), the degree of stretching is more pronounced.

(2) The mean-square amplitude of the Rouse modes $\langle X_{p}\left(0\right)X_{p}\left(0\right)\rangle$ deviates from the model prediction, at least for p>3.

(3) In one of the papers [16], where a cross-correlation function $\langle X_{p}\left(0\right)X_{q}\left(t\right)\rangle$, p≠q, was evaluated, the cross-correlation function at intermediate times became large. Other authors have not tested this result. The actual forms of the bead displacement distribution functions and of the mode cross-correlation functions $\langle X_{p}\left(0\right)X_{q}\left(t\right)\rangle$ appear worthy of further study.

(4) Contrary to the Rouse model, under shear, the response of a chain is to rotate, not to distort affinely without rotation [18].

(5) According to the Rouse model, bead displacements are driven by independent Gaussian random processes. As a result, $g^{\left(1s\right)}\left(q,t\right)$ should be accurately described by the Gaussian approximation. Doob’s theorem [89] then guarantees that $g^{\left(1s\right)}\left(q,t\right)$ decays as a single exponential in time. These predictions are incorrect for polymer coils in the melt. Therefore, bead displacements are not described by independent Gaussian random processes, contrary to the basic assumptions of the Rouse model.

Contrariwise, the simulations reported here give some guidance as to the valid predictions of useful physical models for the dynamics of polymeric fluids at different concentrations. As notable predictions, for short as well as long polymer chains, chain-crossing constraints substantially modify polymer dynamics. The mean-square Rouse amplitudes $\langle {\left(X_{p}\left(0\right)\right)}^{2}\rangle$ scale as $p^{-2}$ for smaller *p*, generally p≤3 but perhaps as $p^{-3}$ for larger *p*. $\langle X_{p}\left(0\right)X_{p}\left(t\right)\rangle$ depends on time as a stretched exponential $\exp(-\alpha t^{\beta })$, in which the stretching exponent β depends on *p*, being larger at the extrema p≈1 and p≈N, and smaller in the middle. A nominal Rouse relaxation time $\tau_{p}$ for the Rouse temporal autocorrelation functions scales as ${\left(N/p\right)}^{x}$, where *x* in different systems is in the range 2≤x≤3.5 depending on *p*, temperature, and other variables. Mean-square center-of-mass motions are subdiffusive at shorter times *t* but diffusive at a sufficiently long time. Correspondingly, the Gaussian approximation $g^{\left(1\right)}\left(q,t\right)∼\exp\left(-q^{2}\langle {\left(\delta R\right)}^{2}\rangle \right)$ is invalid for polymers in melts. By inference, the distribution of bead displacements is not a Gaussian random process. The non-Rouseian behavior of the Rouse modes is greatly reduced if chain-crossing constraints are removed, even for very short chains. This final point suggests a mechanism for the non-Rouseian behavior, namely, if there are density fluctuations adjacent to the chain path, chain-crossing constraints prevent the fluctuations from being relaxed via the diffusion of nearby chains in directions transverse to the chain path, if their paths would pass through the position of the chain of interest, thus slowing the relaxation of density fluctuations.

The language of the Rouse model, the description of chain motion in terms of Rouse modes, is extensively used in polymer simulations. If the Rouse description of chain motions is incorrect, one reasonably asks what alternatives exist to the Rouse modes. If one accepts the qualitative approximation that the local motions of remote parts of a polymer are only weakly correlated, one is led toward coordinates that refer to the motions of local groups of atoms and that accommodate systematic coarse-graining to describe motions of larger and larger sections of a polymer chain. As discussed in our previous paper [90], such coordinates are given by wavelet decompositions [91]. The wavelets first described by Haar [92] appear to provide a natural and intuitive description of segmental motions.

Having noted at the start of this review the Kirkwood–Riseman model, one legitimately asks to what extent the articles reviewed here speak to the validity of this model. In short, predictions from the Kirkwood–Riseman model do not overlap well with the results here. To test the validity of the Kirkwood–Riseman model, it would be more effective to review simulations of polymer melts and solutions in a shear field, which would require a separate article. The Kirkwood–Riseman model refers to an isolated chain in a shear field. Analytical–theoretical extensions of the model to treat non-dilute polymers have been made [7,8] and could be extended further.


*Nonetheless, as our primary conclusion, the above direct tests of the Rouse model correspond to the Rouse model’s major predictions for Rouse mode amplitudes and relaxations. The predictions are rejected by simulations.*



*There can be no doubt that the Rouse model is invalid in polymeric fluids, where it is extensively used as a basis for theoretical models.*


## Figures and Tables

**Figure 2 polymers-15-02615-f002:**
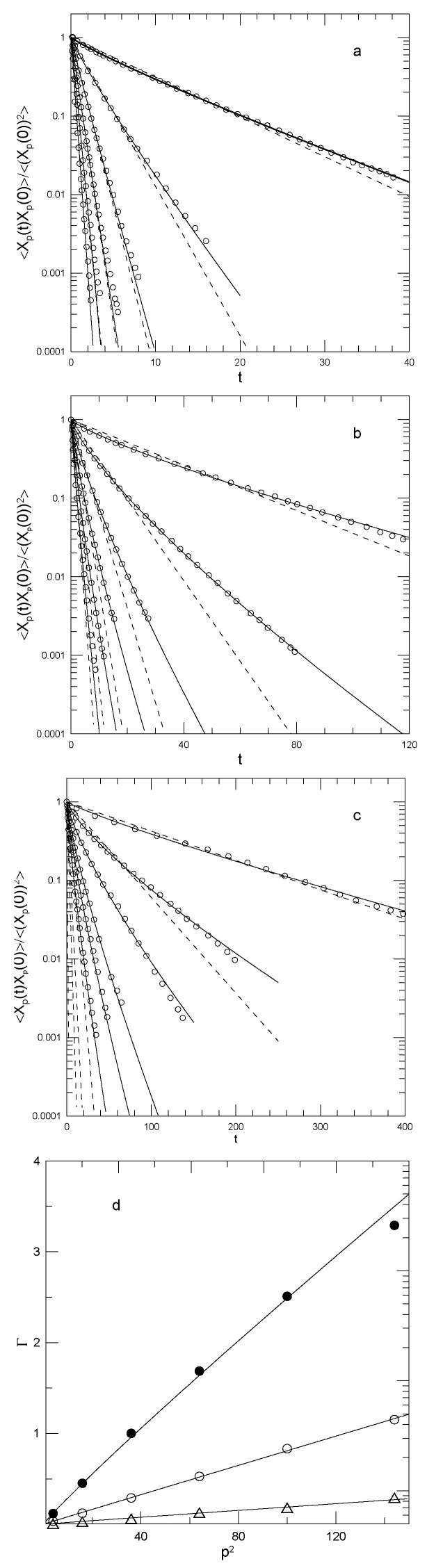
Time autocorrelation functions for Rouse mode amplitudes $X_{p}\left(t\right)$ for p=2,4,6,8,10,12 (top to bottom) for melts of (**a**) 5 kDa, (**b**) 10 kDa, and (**c**) 20 kDa poly(ethylene oxide) ring polymers. Circles represent samplings of points from the original results of Tsalikis et al. [13]; dashed lines are best fits to a single exponential; and solid lines are non-linear least-square fits to stretched exponential fits. (**d**) Γ against $p^{2}$. Lines in (**d**) are power-law fits to $\Gamma =Ap^{2x}$ for x=0.94, 0.99, and 0.98 (top to bottom).

**Figure 3 polymers-15-02615-f003:**
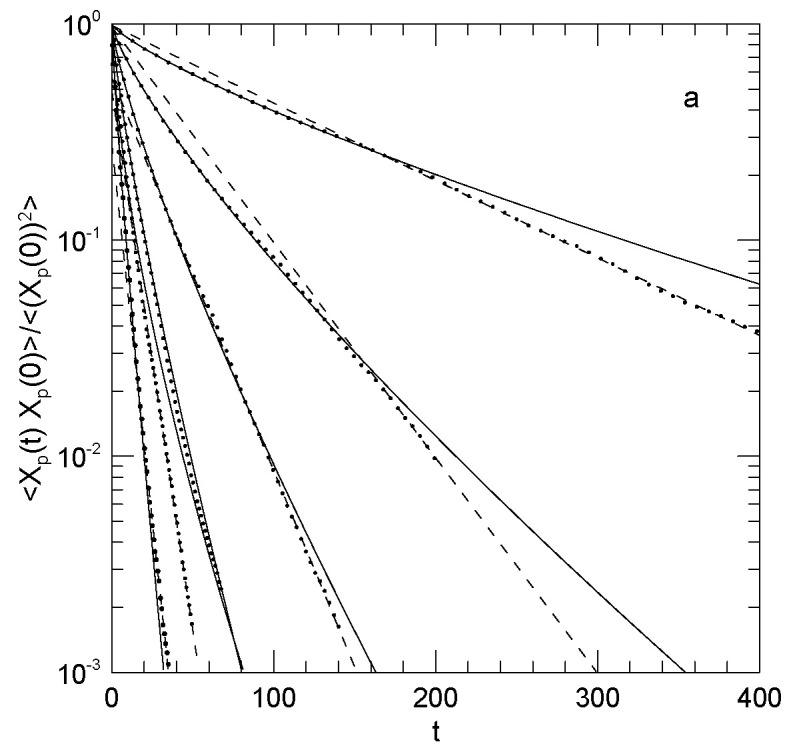
Time autocorrelation functions $\langle X_{p}\left(0\right)X_{p}\left(t\right)\rangle$ of Rouse mode amplitudes $X_{p}\left(t\right)$ for p=2,4,6,8,10,12 (top to bottom) as normalized by $\langle {\left(X_{p}\left(0\right)\right)}^{2}\rangle$ for (**a**) a 20 kDa polyethylene oxide ring melt, and dilute solutions of 20 kDa rings in melts of (**b**) 1.8 kDa, (**c**) 10 kDa, and (**d**) 20 kDa linear polyethylene oxides. Lines of dots are from the original simulations of Papadopoulos et al. ([40], their Figure 4); solid lines represent stretched exponential fits, and dashed lines represent simple exponentials.

**Table 1 polymers-15-02615-t001:** Fitting parameters α and β from $\langle X_{pi}\left(t\right)X_{pi}\left(0\right)\rangle /\langle {\left(X_{pi}\left(0\right)\right)}^{2}\rangle =\exp\left(-\alpha t^{\beta }\right)$, using results of Kremer and Grest [37], their Figure 13, of N = 20, 50, 100, and 200 bead chains in their melts. In some cases, marked * β was fixed. ** fit was only to the points with t<5.

*N*	20	20	50	50	100	100	200	200
*p*	α	β	α	β	α	β	α	β
1	0.66	0.89	0.91	0.90	0.27	0.68	0.120	0.58
2	2.45	0.88 *	3.10	0.75	0.27	0.68	0.27	0.65
3	5.18	0.87	6.16	0.79	1.72	0.75 *	0.49	0.68
4	7.66	0.81	10.26	0.84	3.00	0.7 *	0.68	0.71
5	16.63	0.95	15.16	0.83	3.00	0.7 *	1.00	0.83 **
6					3.00	0.7 *	1.34	0.71*
8							2.23	0.69
10							3.05	0.75 *
20							13.65	0.85 *

**Table 2 polymers-15-02615-t002:** Fitting parameters α and β and average decay rates $\bar{\gamma }$ from $\langle X_{pi}\left(t\right)X_{pi}\left(0\right)\rangle$ based on simulations of Tsalikis et al. [13] of 5, 10, and 20 kDa polyethylene oxide ring melts. $\gamma_{1}$ is the initial slope, corresponding to a simple exponential relaxation.

*p*	*M*(kDa)	$\gamma_{1}$	α	β	$\bar{\gamma }$
2	5	0.117	0.157	0.894	0.119
4	5	0.437	0.513	0.898	0.452
6	5	0.988	1.019	0.958	1.001
8	5	1.675	1.683	0.978	1.687
10	5	2.524	2.527	1.014	2.510
12	5	3.471	4.230	1.297	3.291
2	10	0.033	0.070	0.814	0.0341
4	10	0.118	0.210	0.793	0.123
6	10	0.277	0.403	0.810	0.290
8	10	0.498	0.658	0.807	0.528
10	10	0.777	0.942	0.818	0.833
12	10	1.116	1.196	0.884	1.152
2	20	0.00859	0.0166	0.878	0.00882
4	20	0.028	0.0690	0.786	0.0290
6	20	0.060	0.128	0.782	0.0629
8	20	0.126	0.213	0.804	0.129
10	20	0.181	0.276	0.813	0.184
12	20	0.282	0.404	0.813	0.293

**Table 3 polymers-15-02615-t003:** Fitting parameters for results of Papadopolous et al. [40] for the plots in Figure 3. The parameterization is $\langle X_{p}\left(0\right)X_{p}\left(t\right)\rangle =\langle {\left(X_{p}\left(0\right)\right)}^{2}\rangle \exp\left(-\alpha t^{\beta }\right)$.

Solvent	*p*	$\langle X_{p}{\left(0\right)}^{2}\rangle$	α	β	$\bar{\gamma }$
ring					
	2	1	0.025	0.789	0.008
	4	1	0.063	0.793	0.027
	6	1	0.122	0.791	0.061
	8	1	0.196	0.812	0.120
	10	1	0.389	0.654	0.174
	12	1	0.359	0.852	0.277
1.8 kDa					
	2	1	0.008	1.003	0.008
	4	1	0.042	0.837	0.021
	6	1	0.097	0.812	0.051
	8	1	0.142	0.830	0.086
	10	1	0.263	0.739	0.136
	12	1	0.434	0.679	0.224
10 kDa					
	2	1.015	0.032	0.572	0.0015
	4	0.970	0.0465	0.681	0.0085
	6	0.999	0.120	0.623	0.023
	8	1.28	0.326	0.522	0.063
	10	0.966	0.269	0.653	0.098
	12	1.38	0.538	0.584	0.222
20 kDa					
	2	0.98	0.025	0.720	0.005
	4	1.11	0.135	0.529	0.013
	6	1.03	0.161	0.627	0.038
	8	1.22	0.308	0.570	0.078
	10	1.30	0.466	0.581	0.171
	12	1.16	0.514	0.590	0.210

## Data Availability

All data are in the main body of the paper.

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
