# Peer review of "Simulational Tests of the Rouse Model"

_polymers, 2023, doi:10.3390/polym15122615_

Round 1

Reviewer 1 Report

The author provides a clear summary of the Rouse and Kirkwood-Riseman (KR) models before going to the simulation tests of the predictions of the Rouse model for polymer solutions subject to shear. Both models make unphysical assumptions which may be approximately valid under limited sets of circumstances. In particular, the Rouse model ignores polymer rotation when subject to shear, while the Kirkwood-Riseman model does not consider polymer deformations responding to applied fluid shear.

A comprehensive review of the literature of polymer simulations is given and each contribution is meaningfully summarized, and its results are compared to the different assumptions and predictions of Rouse model. Each work is put in proper perspective by the detailed description of the physical model and the level of atomic resolution used in the simulation.

This work would be a very useful addition to the literature of polymer solutions and I would recommend its publications subject to a few minor optional recommended additions.

On page 4 the author states: “As it happens, there is actually a fundamentally different model of polymer dynamics, the Kirkwood-Riseman model with hydrodynamics suppressed, that predicts the same M-dependence, so the inferential test is non-informative.” While agreeing with the statement, a more accurate ending to the sentence would perhaps be: “… same M-dependence, so the inferential test is a necessary, but not sufficient test of the model”.

In first describing the nature of the “bead” in the descriptions of the Rouse and KR models on page 8, it may be useful to the reader if the language of coarse-graining is used. The author mentions coarse-graining latter on when reviewing other simulation studies and it could be useful to set the stage at the beginning when describing the Rouse model.

Given the number of papers summarized in the work, to give a more global overview, would it be useful to prepare a Table where the columns could be various assumptions of the Rouse model, as given for example, on page 4, and the rows would be the different papers reviewed by this manuscript. Each table entry could be a short indication of whether the paper verifies or refutes the given assumption. This could be a short way of showing the overwhelming simulation-based evidence that the assumptions of the Rouse model are not corroborated.

Two brief discussion topics can be added to Conclusions section. First, at the beginning of the manuscript, the Kirkwood-Riseman model was described in some detail. Can the author comment briefly how the simulations summarized in Sec. 4 would be compared to the predictions of the KR model? Second, can the author state the alternative direction/framework he believes researchers should take in describing their simulations and / or experimental results, instead of the Rouse model.  

Author Response

The referee suggestion on page 4 is sound and has been used.  The referee asks on page 8 for a discussion of beads; this discussion has been added.   I appreciate the suggestion of the table, but most papers here did not test most of the predictions, so the table would have a lot of blanks.  I did modify the conclusions to emphasize the few papers that did validate the model, as opposed to the larger number of papers that did not.  The referee asks how these simulations test the Kirkwood-Riseman model; these fine simulations are not very helpful in this regard, and remarks on this are appended to the discussion.  Finally, the referee asks for alternatives to Rouse coordinates.  I added as a suggestion the use of Haar wavelets, which can be viewed as a systematic and complete mathematical replacement for the approximate idea of the Kuhn length.

Reviewer 2 Report

It has been known for many decades that the Rouse model does not work for polymer melts. So, why should a review paper whose main conclusion is that the Rouse model does not work for polymer melts be of any interest and usefulness?

There is no reason to publish such a review

language 

Author Response

I believe that the referee's claim is refuted by the first paragraph of my paper, and references therein, not to mention standard textbooks, e.g, the recent one from Rubenstein and Colby.  The referee may be referring to the molecular weight dependence of high-molecular-weight polymers, where the Rouse prediction is wrong, but that is hardly the only use of the Rouse model.

Reviewer 3 Report

The article by George David Joseph Phillies is a revised version of the author's preprint and is devoted to review of literature simulations of polymer melts dynamics. The author has analyzed many research articles, which test the Rouse polymer chain model in melts, and gave a general conclusion that this model is invalid.

In general, the article is well written and contains a lot of useful information. I believe that it will be interesting to many readers involved in the modeling of polymer melts and thus the article can be recommended for publication. Before it, it would be nice if the author carefully rereads the manuscript and corrects a number of typos.

A list of typos noted is given below.

1. On Page 10, at the left side of Table 1. “!b” looks like typos.

2. On Page 10, at bottom. In the sentence “Static properties that were examined include the mean-square” “end-to-end distance, the radius of gyration,” There is a line break here.

3. On Page 11, in Figure 1c. The range of values on the vertical axis would be better done as in Figure 1a,b, and d.

4. On Page 12, in Figure 2a,b,c. It looks like that indexes in the autocorrelation functions on the vertical axis labels should be the same (not p and q).

5. On Page 13, paragraph 4 from the top. The expression <(Xpa(0))^2 is missing an angle bracket.

6. On Page 23, paragraph 2 from the top. The author writes “In this Chapter”. Perhaps the author means "Section".

7. On Page 30, paragraph 4 from the top. The expression <Xp(t)Xp(0) is missing an angle bracket.

8. On Page 30, line 3 from the bottom. The expression âŸ¨Xp(t)X)q(0)⟩ has an extra parenthesis.

9. On Page 31, paragraph 1 and 2 from the top. The expressions <Xp(t)Xp(0) are missing an angle bracket.

10. On Page 31, paragraph 2 from the top. The same typo.

Author Response

I am most grateful for the list of typos and figure corrections, and have made the requested corrections.

Reviewer 4 Report

The paper offers a detailed critical analysis of Rouse model as a candidate for the description of dynamics of model polymer chains. While it is generally accepted that the model fails, the review paper provides a large body of very detailed evidence confirming the deficiency of Roussian approach. A strong point of the review is an interesting discussion of various aspects of non-standard diffusion found in model polymers. Those results can offer a starting point for revisions/modifications of Rouse model.

Recommendation: the abstract should be rewritten - it is a rather technical summary of the paper and not a kind of introduction.

Author Response

I apologize to the referee, but I was brought up on the idea that the abstract is a technical summary.  I have tried to rewrite the abstract, to give a qualitative discussion first, but may have failed to satisfy the referee.